# A neural correlate of perceptual segmentation in macaque middle temporal cortical area

Andrew M. Clark [1,2] ✉ & David C. Bradley[1]

High-resolution vision requires fine retinal sampling followed by integration to recover object properties. Importantly, accuracy is lost if local samples from different objects are intermixed. Thus, segmentation, grouping of image regions for separate processing, is crucial for perception. Previous work has used bi-stable plaid patterns, which can be perceived as either a single or multiple moving surfaces, to study this process. Here, we report a relationship between activity in a mid-level site in the primate visual pathways and segmentation judgments. Specifically, we find that direction selective middle temporal neurons are sensitive to texturing cues used to bias the perception of bi-stable plaids and exhibit a significant trial-by-trial correlation with subjective perception of a constant stimulus. This correlation is greater in units that signal global motion in patterns with multiple local orientations. Thus, we conclude the middle temporal area contains a signal for segmenting complex scenes into constituent objects and surfaces.

Vision relies not only on the fine discrimination of elemental image features such as edge orientation and velocity, but, critically, on the appropriate integration of these features to compute environmental properties such as object shape and trajectory[1]. Problems arise, however, when the retinal image supports more than one equally plausible feature grouping[2–4] (Fig. 1a). For example, when two groups of velocity signals occur in close proximity, the interpretation could reasonably be that of one moving object or several[5] (Fig. 1b). This illustrates the subjective nature of segmentation, i.e., it is not a fixed property of the image, but rather an interpretive process. Despite its obvious importance for normal perception, our understanding of the neural substrates of perceptual segmentation remains at best incomplete.

Visual motion processing has been well characterized and thus provides an excellent model for studying the neural circuits underlying perceptual segmentation[6]. Several computational studies have noted the utility of a two-stage model of motion processing in which an initial high-resolution estimate is followed by a selective integration of local samples to smooth out noise and recover object velocities[7,8]. Importantly, the visual system must take care to limit this integration to only those local samples that arise from a common object. Psychophysical

studies have delineated the physical factors that influence how local motion signals are segmented[9–14], but both the anatomical locus and the form of the neural code remain open questions. Multiple reports have implicated direction selective cells in the primate middle temporal (MT) cortical area as a candidate neural substrate[15–27].

Importantly, in these previous experiments, changes in neural activity were correlated with physical changes in a visual stimulus. However, as noted above, segmentation is fundamentally a perceptual process; therefore, studying its neural substrates requires relating variations in neural activity to variations in the perception of a fixed stimulus. Accordingly, we trained two macaque monkeys to report whether perceptually bi-stable plaid patterns, formed by superimposing drifting square-wave gratings, appeared as a single surface or two independent surfaces. To examine the relationship between neural activity and segmentation judgments, we recorded single-unit activity in MT while monkeys performed this task.

We found a significant trial-by-trial correlation between MT activity and perception. This correlation was present whether or not a stimulus contained explicit segmentation cues. Moreover, the strength of this effect was correlated with both sensitivity to segmentation cues

[1]Integrative Neuroscience Program, Department of Psychology, University of Chicago, Chicago, IL 60637, US. [2]Present address: Department of Ophthalmology and Visual Science, Moran Eye Center, University of Utah Hospitals, University of Utah, Salt Lake City, UT 84132, US. ✉e-mail: clark.andrew@utah.edu

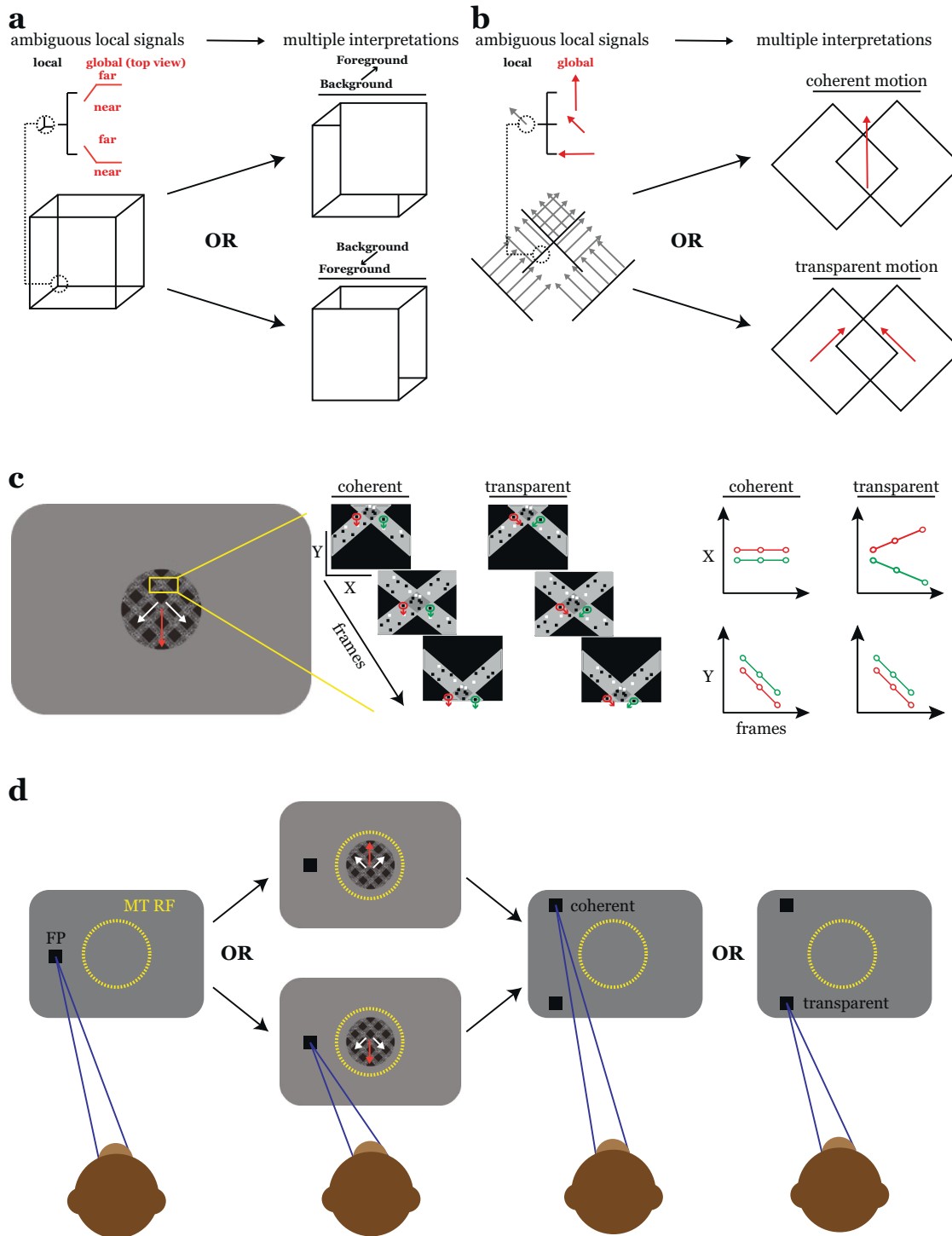

as well as pattern index. The latter quantifies the degree to which a unit signals global rather than local motion in complex patterns[21]. Although pattern direction selectivity has long been considered a defining feature of MT, and pattern direction selective cells exhibit tuning to complex stimuli that tracks with human perception of these stimuli[19], to our knowledge this is the first demonstration of a correlation between pattern index and perceptual segmentation judgments.

## Results

### Monkeys use texturing cues to segment plaid motion and perceive plaids as either coherent or transparent motion

We trained two monkeys to indicate their perception (coherent or transparent motion) of drifting plaid stimuli. Human observers typically perceive these stimuli as either coherent or transparent motion with roughly equal frequency. In order to provide a correct answer on a given trial, and assign a basis for operant reward, we created a segmentation cue by texturing the component gratings that formed the plaid (Fig. 1c, d). In the coherent condition, all texture moved in the pattern direction (Fig. 1c, "coherent"). In the transparent condition, the texture moved perpendicular to the orientation of the grating on which it was superimposed (Fig. 1c, "transparent"). We controlled task difficulty by varying the contrast of this texture cue. On cued trials, monkeys were rewarded for a response congruent with the texture cue, rewards were delivered randomly (50/50 odds) on trials that contained patterns without a texture cue (zero texture contrast condition).

**Fig. 1 | The hard problem of segmentation in visual perception. a** Cartoon illustration of the problem of perceptual segmentation. An observer's perception of depth in the Necker cube (left) alternates between two plausible interpretations (right). This is because there are no cues in the image that allow the brain to determine unambiguously the three-dimensional orientation of the figure (provided by the monocular cue of occlusion on the right). **b** When presented with multiple motion signals in close spatial proximity the visual system must determine whether local samples arise from one or multiple objects. The ambiguity inherent in local motion signals, that is, a family of object motions can yield the same local motion, results in multiple equally plausible interpretations of the visual input, i.e., the vector fields here could arise from the coherent motion of a single surface or the transparent motion of overlapping surfaces. **c** (left) Example of our textured plaid stimuli. Square wave gratings drifting normal to their orientation ("component directions" - white arrows) were superimposed to form plaid patterns. Plaids could be perceived as either coherent motion in a single, pattern, direction (red

arrow) or transparent motion in the component directions. Plaid perception was biased through the addition of a random dot texturing cue. (middle) The region highlighted in yellow is expanded and shown for a sequence of frames separately for coherent and transparent cues. Dot motion in each case is represented by the green and red arrows. (right) The (x,y) positions of the highlighted dots are plotted versus frame number. In the coherent case, all texture drifted in a single direction. In the transparent case, texture moved in the component directions. **d** Cartoon illustration of our motion segmentation task. Monkeys began each trial by fixating on a small point. After a brief delay, a plaid pattern with a particular type (coherent/transparent) and magnitude (e.g., contrast) of texture cue appeared at the location of the MT RF. Plaids could drift in one of two possible pattern directions on each trial. After stimulus offset, choice targets appeared well above and below the MT RF. Monkeys needed to indicate their plaid perception via a saccade to the appropriate choice target.

Behavioral data from two representative experiments is displayed in Fig. 2a, responses are plotted as the proportion of coherent judgments versus the contrast of the texture cue (by definition, transparent contrasts assume negative values), separately for patterns drifting either up or down. Overall, the monkeys' perception of coherence/transparency was reliably affected by both the sign (transparent, coherent) and strength (contrast) of the texture cue (ANOVA; monkey N: direction – $F = 0.58$, $p = 0.45$, sign – $F = 1248$, $p < 10^{-10}$, contrast – $F = 22.63$, $p < 10^{-10}$; monkey S: direction – $F = 0.41$, $p = 0.52$, sign – $F = 2876.7$, $p < 10^{-10}$, contrast – $F = 36.5$, $p < 10^{-10}$). Cumulative Gaussian functions were fit to the data from each session to characterize the monkeys' psychophysical performance. The distribution of the goodness-of-fit of these models for both monkeys across all sessions is given in Fig. 2b. Overall, monkeys performed accurately and consistently on the task, we rejected fewer than 13% of sessions across both monkeys for poor fits of the cumulative Gaussian model.

As noted above, both the contrast of the texture cue and the direction of pattern motion were varied across trials, with stimuli drifting either upward or downward on a given trial. This was done to minimize both psychophysical[11] and neuronal[28] adaptation effects. There was no significant effect of pattern direction on either the offset (point of subjective equality or PSE) (Wilcoxon rank-sum test; monkey N: $z = 0.25$, $p = 0.8$; monkey S: $z = 0.86$, $p = 0.39$) or threshold (Wilcoxon rank sum; monkey N: $z = 0.14$, $p = 0.89$; monkey S: $z = 0.49$, $p = 0.62$) of fitted functions (Fig. 2c). Furthermore, there was no significant difference between monkeys in the magnitude of texture contrast necessary to support a threshold level of performance (monkey N = 24.5% ± 3.9%, Monkey S = 18.9% ± 1.9%; Wilcoxon rank-sum, $z = 1.01$, $p = 0.31$).

Across sessions, we varied the inter-grating angle separating component-grating directions. Psychophysical studies have shown that humans are more likely to perceive plaids as coherent when this angle is smaller[10]. If the monkeys were faithfully reporting their perception of coherence/transparency, then, based on these findings, the PSE, that is, the texture contrast corresponding to an even split of coherent and transparent choices, would be expected to increase with increases in inter-grating angle. This was indeed the case (Fig. 2d; collapsing across pattern directions, Kruskal–Wallis; monkey N: $\chi^2 = 23.06$, $p < 10^{-3}$; monkey S: $\chi^2 = 22.22$, $p < 10^{-3}$; correlation between normalized inter-grating angle and PSE - monkey N: $r = 0.67$, $p < 10^{-9}$; monkey S: $r = 0.76$, $p < 10^{-13}$). In contrast, varying the inter-grating angle had no significant effect on the slope of the psychometric function (Fig. 2d; collapsing across pattern directions, Kruskal–Wallis; monkey N: $\chi^2 = 8.09$, $p = 0.23$; monkey S $\chi^2 = 3.18$, $p = 0.67$; correlation between normalized inter-grating angle and slope – monkey N: $r = -0.4$, $p = 0.2$; monkey S: $r = 0.03$, $p = 0.76$). Thus, consistent with human psychophysical data, the average effect of varying the inter-grating angle was a shifting of the bias point, rather than an increase or decrease in sensitivity to the segmentation cue.

Finally, rewards were dispensed randomly with a probability of 0.5 on zero-texture contrast trials. If monkeys were both aware of this unique contingency and were able to distinguish zero texture contrast from cued stimuli, then they might have developed distinct strategies for these two types of trials. Two observations strongly suggest that this did not occur. First, there was a qualitatively similar effect of varying the inter-grating angle on judgments of both cued and zero texture contrast plaids (Fig. 2d and Supplementary Fig. 1). Second, for both monkeys, choices on bi-stable trials were not significantly more likely to be repetitions of recently (one trial previous) rewarded choices (binomial test, monkey N: 0.52, $z = 0.74$, $p = 0.22$; monkey S: 0.51, $z = 0.9$, $p = 0.18$).

In conclusion, monkeys' performance on our segmentation task was under good stimulus control. The dependence of perceptual judgments on the sign and magnitude of the texture cue and the variation in the PSE with changes in inter-grating angle suggest that monkeys were reporting their subjective perception of motion coherence/transparency. Finally, monkeys' responses on zero texture contrast trials were unaffected by reward history on the previous trial and were significantly affected by changes in inter-grating angle. This suggests that monkeys continued to report their subjective perception of plaid surface configuration in this important condition.

## MT neurons signal surface configuration in coherent and transparent plaid patterns

As detailed above, the progression of texture contrast from negative to positive equates to a perceptual transformation of the stimuli from transparent to coherent. In general, for a given MT unit, responses tended either to rise or fall as the texture contrast went from negative to positive, with the direction of this effect often depending upon the directions of pattern/component motion. As an example, the direction tuning curves of two representative MT units are shown along with these units' responses to plaids containing either low or high contrast coherent or transparent texture cues in Fig. 3. We sought to better quantify these plaid responses in a manner that could be related to the psychophysical performance of our monkeys.

To examine the relationship between plaid surface configuration (coherent or transparent), as signaled by our texture cue, and MT activity, we first classified cells in terms of their preference for coherent motion (positive slope) or transparent motion (negative slope) by regressing firing rates on signed cue contrast (separately for each pattern direction). Examples of these plaid tuning curves for the same example units from Fig. 3 are shown in Fig. 4a. Following classification, we used receiver operating characteristic (ROC) analysis to quantify each unit's sensitivity to modulation of the texture cue (see Methods). Neurometric functions obtained in this manner could then be directly compared to the monkeys' psychophysical performance in the same session, to directly compare neuronal to psychophysical sensitivity to plaid texture. We carried out this signal detection analysis twice for all

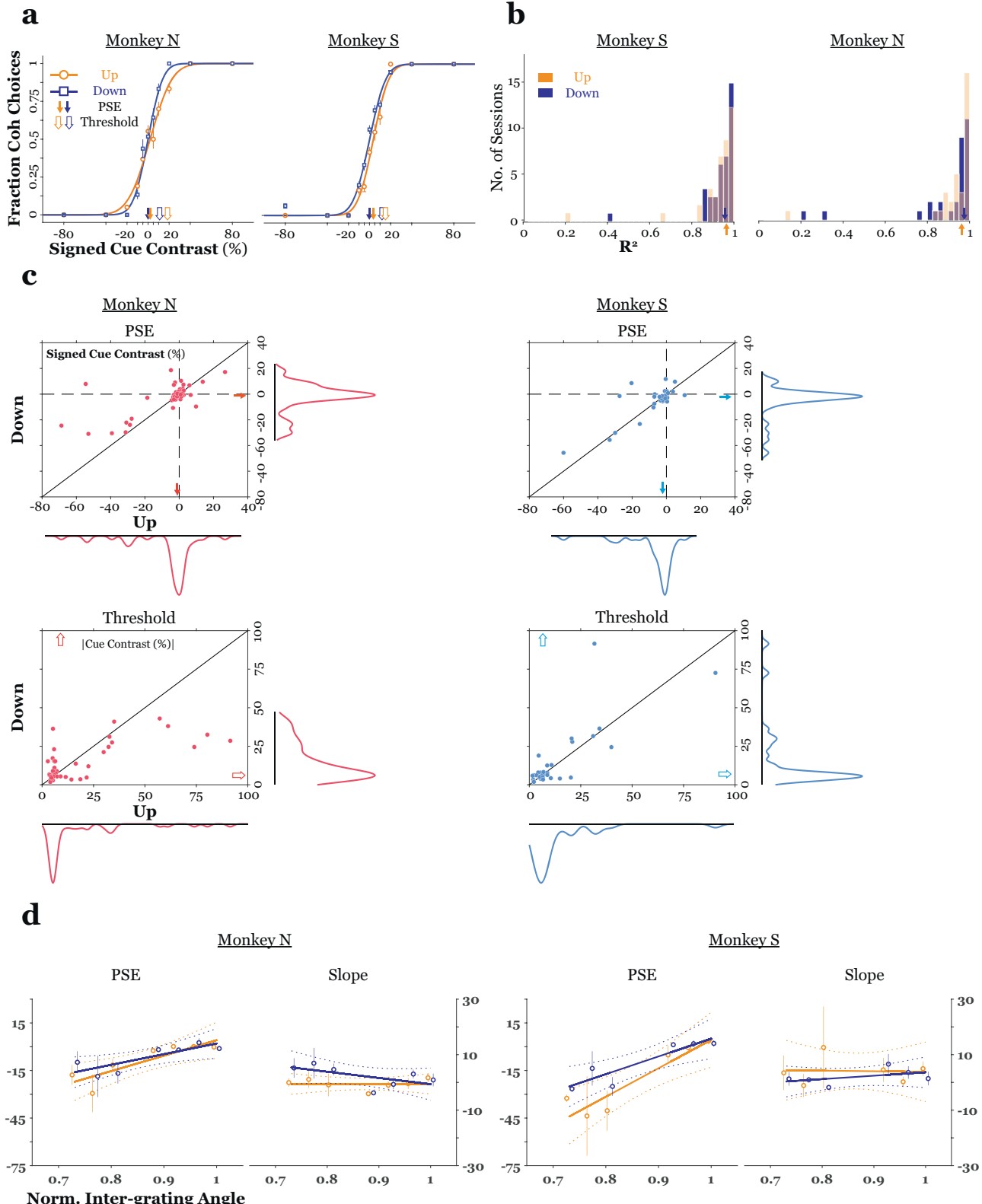

units in our sample, computing separate neurometric functions for each direction of pattern motion (again, up or down). Importantly, for this analysis, we only included trials in which: (i) the stimulus contained a texture cue, and (ii) the monkeys' response was congruent with this cue (i.e., "correct" trials).

Plaid tuning curves and neurometric functions for the two representative MT units and the associated psychometric functions collected along with these responses, are shown in the top and bottom panels, respectively, of Fig. 4a, b. These units showed a roughly monotonic increase or decrease in response with transformation of the texture cue from transparent to coherent. Furthermore, the direction and strength of this relationship depended upon the directions of plaid motion. Finally, the neurometric functions calculated from these cells' responses only approached (but still fell short of) psychophysical performance for a single direction of plaid motion. Both the neurometric and psychometric functions were summarized

**Fig. 2 | Performance in a motion segmentation task. a** Examples of monkeys' behavior in a representative session ($n \geq 20$ trials per stimulus condition). In the left (right) panel, data from a single session from monkey N (S) is plotted as the fraction of coherent choices (ordinate) versus the signed contrast of the texture cue (abscissa). Here, transparent (coherent) texture assumed negative (positive) values. Responses are plotted separately according to the direction of pattern motion on a trial (up (90°) or down (270°)). For both animals, performance, either the contrast for which answers split 50/50 (PSE – filled arrows) or the amount of texture contrast required to support a specific level of performance (threshold – open arrows), was similar across drift directions in these sessions. **b** Histograms of the $R^2$ values for fitted cumulative Gaussian functions. Data from monkey S (N) are shown at the left (right). **c** (top) PSE measured for plaids drifting down (ordinate) is plotted versus PSE for plaids drifting up (abscissa), marginals represent the PSE distribution for each condition, arrows mark the mean for each condition. Data for all sessions from monkey N (S) is given in the left (right) column. (bottom) Same conventions as for PSE data, but for the threshold of fitted functions. There was no significant difference in either PSE or threshold across pattern directions (see text). **d** PSE and slope (ordinates) are plotted versus the normalized angle separating component grating directions ("inter-grating angle" - abscissa). Open circles are means; solid line is the best-fit regression model, dashed lines are 95% confidence intervals for the regression models. There was a significant correlation between PSE and normalized inter-grating angle but not between slope and normalized inter-grating angle, suggesting a shifting, but not steepening or flattening, of the psychometric function with changes in the angle separating component gratings. (monkey N, $n = 32$ sessions; monkey S, $n = 43$ sessions). In all panels, error bars represent standard error of the mean.; coh. coherent, PSE point of subjective equality, Norm. normalized.

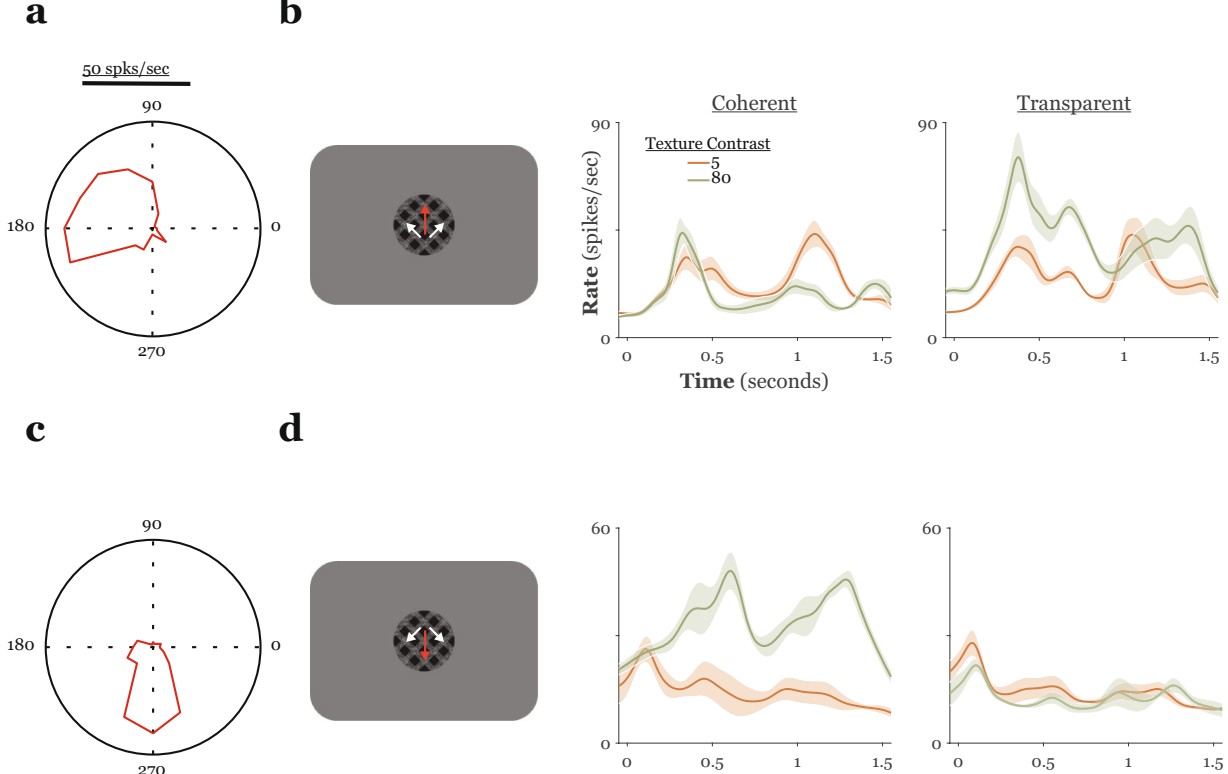

**Fig. 3 | Representative MT single-unit responses to textured plaid stimuli.**
**a** Polar plot of the direction tuning profile in response to a single sine grating for a representative MT unit from monkey S. The angle represents direction of grating motion and magnitude represents firing rate, this unit's preferred direction roughly overlapped one of the component directions in plaids with a pattern direction of 90° (up). **b** Peri-stimulus time histograms (PSTHs) in response to plaids drifting in a pattern direction of 90° (shown schematically on the left) for the unit shown in **a**. Responses are sorted according to the type (coherent/transparent – middle/right panels, respectively) and Michelson contrast (color cued across PSTHs) of the texture cue. Only responses on correct trials for a low and a high contrast texture cue of each type are shown. This cell responded better to upward drifting plaids with transparent texture cues, with responses to these patterns increasing with increasing texture contrast. **c**, **d** Conventions as in **a** and **b** but for a different MT unit from monkey S with a preferred direction that nearly overlay the pattern direction for downward drifting plaids. This unit preferred downward drifting plaids with coherent texture cues, with responses to these patterns increasing with increasing texture contrast. In all panels, shaded regions represent the standard error of the mean. spks. spikes, sec. seconds.

in terms of threshold, that is, the contrast corresponding to ~84% correct choices (corresponding to the mean + 1 s.d. of the fitted cumulative Gaussian function). Over the sample, the N/P ratio, the ratio of neurometric to psychometric threshold, averaged $12.4 \pm 1.2$ in monkey N and $15.9 \pm 1.8$ in monkey S and was between 0.5 and 1.5 for at least one direction of plaid motion in only ~16% (18%) of units from monkey N (monkey S) (Fig. 5a). From the example units shown in Figs. 3 and 4 it appears that neuronal sensitivity might have been affected by the relationship between a unit's preferred direction and the direction(s) of plaid motion used in the experiments. Specifically, the direction tuning curves in Fig. 3a, c suggest a relationship between a neuron's direction tuning for single sine gratings and its sensitivity

transparent/coherent motion in our textured plaids. This was the case for both monkeys (ANOVA; relative preferred directions binned with 10° resolution; monkey N: $F = 2.12$, $p < 0.01$; monkey S: $F = 2.01$, $p < 0.01$). Given the large degree of variability in neuronal sensitivity (Fig. 5a), to visualize the dependence of neuronal sensitivity on relative preferred direction, we first normalized each unit's preferred direction to the "best" direction of plaid pattern motion (that is, the plaid direction that yielded the smallest angle between a unit's preferred direction and the plaid pattern direction). We found that relative neuronal threshold (threshold for the "worst" plaid direction/threshold for the "best" plaid direction) varied with this normalized preferred direction, with peaks in this threshold ratio occurring around either

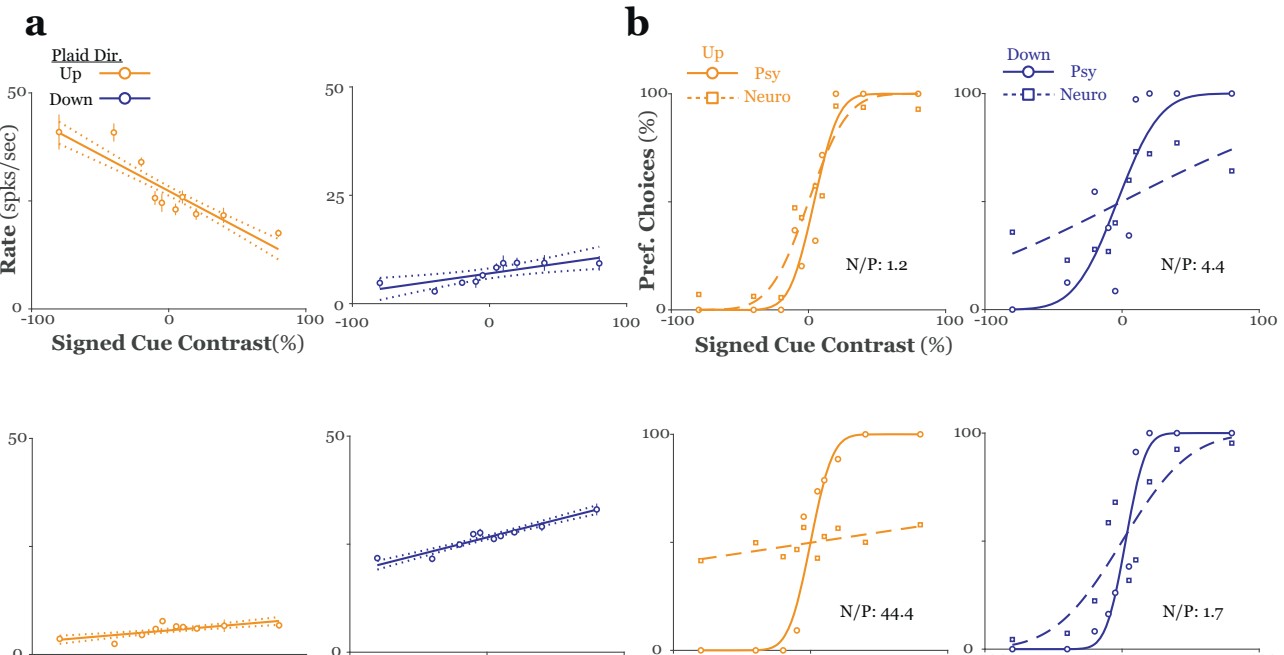

**Fig. 4 | Quantifying MT responses to textured plaids. a** Firing rate is plotted against signed texture contrast separately for plaids drifting either up (left) or down (right), solid lines are best fit linear regressions, data in the top (bottom) rows are from the unit shown in Fig. 3a, b (Fig. 3c, d). The sign of the regression slope was used to assign a preferred texture cue (coherent/transparent) for each unit/plaid direction combination ($n \geq 20$ trials for each stimulus condition). Error bars represent standard error of the mean. **b** Neurometric functions for the units shown in **a** are depicted alongside psychometric functions collected during the same session. For each function, we now plot the percentage of *preferred* cue choices (ordinate) (see text) against signed texture contrast (abscissa). Texture contrast was reordered so that preferred cues assume positive values and null cues assume negative values. Data from upward (downward) drifting plaids is shown in the left (right) panels; data in the top (bottom) row are from the unit shown in Fig. 3a, b (Fig. 3c, d). The ratio of neurometric to psychometric threshold (N/P) is given in each panel. spks. spikes, sec. seconds, dir. direction, pref. preferred, psy. psychometric, neuro. neurometric.

the pattern or component directions (Fig. 5b). This effect could not be explained by a bias in the distribution of preferred directions in the units in each sample towards one of the plaid pattern or component directions (Fig. 5c; Rayleigh test; monkey N: $z = 8.33$, $p < 10^{-3}$, circular mean = 190.13 deg ± 9.83 deg; monkey S: $z = 0.79$, $p = 0.45$) and was consistent across plaid inter-grating angles (Supplementary Fig. 2). Thus, neuronal sensitivity to our textured plaids depends, at least to some extent, on a fundamental tuning property in MT.

In summary, MT responses were modulated by both the direction of plaid motion and the specifics of our segmentation (texture) cue. A comparison of neuronal and psychophysical sensitivity revealed that, in general, MT units were much less sensitive than the monkeys' to the contrast of the texture cue. However, neuronal sensitivity did vary with the difference between a unit's preferred direction and plaid motion direction. The most sensitive cells tended to have direction preferences that nearly overlay either the plaid pattern or one of the component directions, and a small fraction of our sample was as or more sensitive than the monkeys' perception to differences in cue contrast. To determine if signals from these more sensitive units bore a greater relationship with the monkeys' perception, we examined the trial-by-trial correlation between perception and neuronal responses.

## MT activity co-varies with trial-by-trial judgments of plaid surface configuration

An important step in determining a link between neural activity and behavior is establishing a trial-by-trial correlation between the neuronal and behavioral responses to a constant stimulus[29]. To relate neural responses to segmentation judgments, it is crucial, then, to create a stimulus that, despite its invariance, is perceived differently from trial to trial. In the present study, this is represented explicitly by the zero texture contrast plaid. Although we stress that, based on the animals'

psychometric functions, plaids with minimal (less than ~20%) texture contrast were often variably perceived as coherent or transparent as well.

To quantify the extent to which MT responses co-varied with perceptual reports we performed a choice probability (CP) analysis of our plaid data (see Methods – examples of single unit responses sorted according to perceptual report are given in Supplementary Fig. 3). Briefly, CP is a non-parametric, criterion free measure that quantifies the association between spiking responses and perceptual judgments[30]. Limiting the analysis to trials involving zero texture contrast plaids, and sessions in which the monkey made at least five choices of each type for these trials, we calculated CP separately for each direction of plaid motion. Across monkeys, we observed a mean CP value significantly greater than we would expect by chance (Fig. 6a, d; monkey N: mean CP: 0.54, 95% CI: (0.53, 0.56), two-sided *t*-test against null of CP = 0.5, $t = 6.7$, $p < 10^{-9}$; monkey S: mean CP: 0.55, 95% CI: (0.54, 0.57), two-sided *t*-test, $t = 9.4$, $p < 10^{-13}$). Thus, MT neurons tended to fire more when the animals' perception of plaid motion matched a cell's preference, even in the absence of any explicit segmentation cues.

Some prior studies have reported a dependence of CP on the relative number of trials in the underlying rate distributions; that is, the measure is less reliable for stimuli that yield large differences in the proportion of each type of choice[30,31]. To examine this effect in our data we calculated CP separately for all stimuli, regardless of signed texture contrast, for which the monkey made at least a single error trial. CP is plotted against choice ratio (pref/null) separately for each animal in Fig. 6b and e (left panel). From the moving average traces, it is apparent that CP remained above chance for a broad range of choice ratios, only decreasing as ratios decreased (increased) below (above) 0.2 (0.8). Based on the animals' psychometric performance, we would

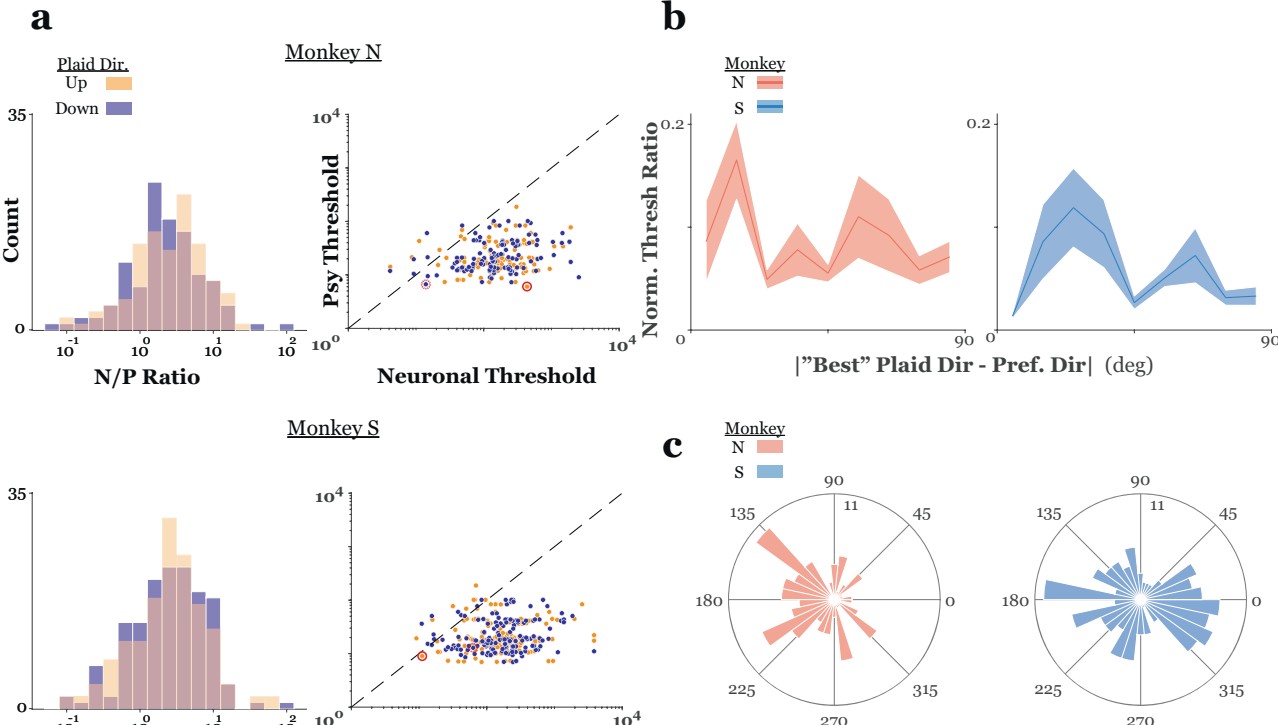

**Fig. 5 | Neuronal sensitivity to plaid segmentation cues. a** The left panels show the distribution of the N/P ratio (neuronal/psychophysical threshold); each cell contributes two data points, one for each direction of pattern motion. In the right panels, psychophysical threshold (ordinate) is plotted versus neuronal threshold (abscissa) for all units in the sample. Data in the top (bottom) row is from monkey N (S). **b** The normalized threshold ratio is plotted against the magnitude of the difference between the best plaid direction and a unit's preferred direction. The "best"

direction was defined as the plaid pattern direction that was closest to a unit's preferred direction (measured with single sine gratings). Data was first binned by normalized preferred direction (10° bins), threshold ratios were then normalized to the maximum and averaged within each bin. Units with preferred directions that were slightly greater or less than plaid component directions showed the greatest difference in sensitivity across plaid pattern directions. **c** Rose histograms of the distribution of preferred directions for all MT units recorded from each monkey.

expect choice ratios of this magnitude only for stimuli with high contrast texture cues (either coherent or transparent) (cf. the example psychometric functions in Fig. 2a, b). To determine whether this was the case, and whether a significant CP persisted even for stimuli with explicit segmentation cues, we examined the effect of absolute value of texture contrast on CP (Fig. 6b, e - right). As expected, CP was significantly greater than chance for stimuli that contained up to moderate (~20% contrast or less) segmentation cues.

In direction, speed and disparity discrimination tasks, MT CPs tend to be greatest in the most sensitive neurons, presumably because these carry the most informative signals[30,32–34]. Consistent with these findings we observed a modest but significant correlation between grand CP, calculated from z-scored firing rates across the texture cue contrasts highlighted in the rightmost panel of Fig. 6b, e, and neuronal threshold (Fig. 6c, f; geometric mean regression; monkey N: $r = -0.12$, $p = 0.07$ monkey S: $r = -0.18$, $p < 10^{-3}$). Thus, signals from the most informative units tended to exhibit a greater co-variation with the monkeys' subjective segmentation judgments, importantly, regardless of any texturing cues added to bias perception.

Given that we had previously determined a relationship between sensitivity to the plaid texture cue and neuronal preferred direction, we wondered whether there was a similar relationship between CP and preferred direction (Fig. 6g). This relationship was only marginally significant in monkey S (ANOVA; monkey N: 1.03, $p = 0.46$; monkey S: $F = 1.73$, $p = 0.04$). In neither animal did we observe a difference in CP across plaid inter-grating angles (Fig. 6h; ANOVA; monkey N: $F = 1.8$, $p = 0.11$; monkey S: $F = 0.32$, $p = 0.9$).

Finally, prior work has demonstrated changes in CP throughout the course of a trial. With some studies reporting a sharp rise and then a relatively flat choice effect[30] and others reporting a continuous

increase in the choice signal throughout a trial[31]. For each monkey, we calculated each unit's CP on zero-texture contrast trials (separately according to pattern direction) in 100 ms bins stepped every 20 ms from just before stimulus onset to just following stimulus offset before averaging. Average CP dynamics for both monkeys are shown in Fig. 6i. In both cases, CP remained at or very near chance levels until almost 500msec after stimulus onset, at which point there was a sharp rise in CP.

### MT pattern direction selectivity correlates with choice probability

In addition to varying with sensitivity, CP has also been shown to be affected by the particular quality of a unit's tuning properties. For example, Uka and DeAngelis[34] found that CP in a binocular disparity discrimination task was dependent upon the symmetry of a unit's binocular disparity tuning curve. In the present case, a related question would be whether sensitivity was greater in pattern direction-selective (PDS) cells than in component direction-selective (CDS) cells. PDS cells encode the overall direction of a pattern containing multiple local orientations, whereas CDS cells respond to the motion of a pattern's oriented components[21] (Fig. 7a).

Thus, in a separate block of trials, we measured responses to sine gratings and plaids to classify the neurons in our sample as PDS or CDS (see Methods). Grating tuning curves, pattern-component predictions constructed from these tuning data, and plaid tuning curves for the units shown in Figs. 3 and 4 and Supplementary Fig. 3 are shown in Fig. 7b. The distribution of pattern and component selectivity, and the preferred directions of the units in each category, are shown separately for each monkey in Fig. 7c and Supplementary Fig. 4, respectively.

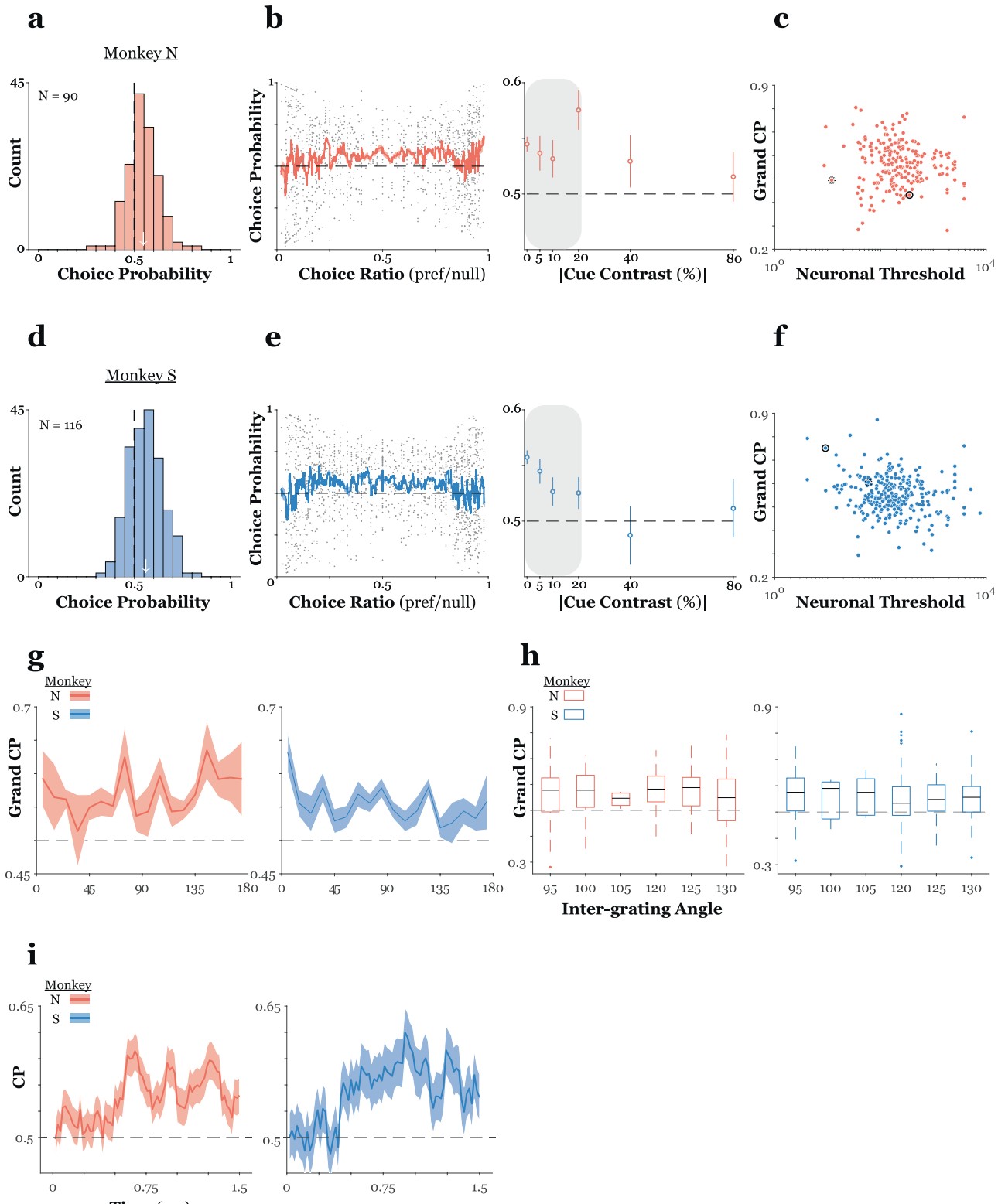

To assess the dependence of CP on pattern-component tuning, we first calculated a pattern index[35] (PI), for which large (small) values indicate greater PDS (CDS)-like behavior. Given the demonstration above that: (i) neuronal sensitivity varied with the difference between a cell's preferred direction and the directions of stimulus motion, and (ii) there was a significant correlation in our sample between neuronal sensitivity and choice probability, we looked at the relationship between PI and grand CP for each unit's "best" direction of pattern motion (see above). We found that CP was significantly correlated with PI (Fig. 7d; geometric mean regression; grand CP monkey N: $r = 0.23$, $p < 0.01$; bi-stable CP monkey N $r = 0.21$, $p = 0.013$; grand CP monkey S: $r = 0.30$, $p < 10^{-4}$; bi-stable CP monkey S: $r = 0.29$, $p < 10^{-3}$), indicating that cells classified as PDS exhibited greater choice-related activity than CDS and unclassified cells. Because both PI and neuronal sensitivity are correlated with CP we performed a multiple regression analysis (with PI and neuronal sensitivity as independent variables and

**Fig. 6 | MT activity co-varies with perceptual segmentation judgments on a trial-by-trial basis. a** Distribution of choice probabilities for plaids with no texture cue for the sample recorded from monkey N. Each cell could contribute up to two data points (one for each direction of plaid motion). A mean CP greater than chance (white arrow) indicates that, overall, there was a significant relationship between MT activity and perception. **b** To examine the influence of any potential choice biases we calculated CP separately for any stimulus for which the monkey made at least a single error. Choice probability is plotted against choice ratio (pref/null) for all stimuli (left) and versus the absolute value of the contrast of the texture cue (right – data from 120 single units). Solid line and the shaded region in the left panel is the mean ± s.e.m. for a 20-point moving average. Choice probabilities calculated for stimuli with unbalanced choice ratios, e.g., those plaids with high cue contrast, were more variable and clustered around chance. Gray shaded region in the right panel highlights the cue contrasts included in the grand choice probability calculation. **c** Grand choice probability (ordinate) is plotted versus neuronal threshold (abscissa). There was a significant negative correlation between choice probability and threshold. **d–f** Convention as in **a–c** but, unless otherwise noted, are for data from 157 single units from monkey S. **g** Grand choice probability (ordinate) is plotted against normalized preferred direction (abscissa) separately for both monkeys. Each MT unit contributed two data points (one for each plaid pattern direction). **h** Box plots of grand choice probability for each inter-grating angle. Solid lines mark the median, the bottom and top edges of the box indicate the 25th and 75th percentiles respectively, whiskers extend to 1.5x the inter-quartile range, outliers are marked beyond this limit. Data in the left (right) panel is from 120 (157) single units from monkey N (S). **i** Grand choice probability (ordinate) is plotted against time from stimulus onset (abscissa). Grand CP was calculated in a sliding bin (100 ms width, 10 ms steps) throughout a trial and then averaged across units.

grand CP as the dependent variable) to rule out the possibility that a correlation between the two measures was responsible for either effect. Both partial correlation coefficients were significant (monkey N: threshold vs. CP: $r = -0.13$, $p = 0.04$, PI vs. CP: $r = 0.23$, $p < 0.01$; monkey S: threshold vs. CP: $r = -0.16$, $p = 0.03$, PI vs CP: 0.29, $p < 10^{-3}$), suggesting that CP increases with sensitivity and in an independent fashion increases with PI.

## Discussion

We recorded single-unit activity from area MT while monkeys reported their perception of patterns that could appear as either coherent or transparent motion. Neuronal sensitivity to segmentation cues added to bias perception varied broadly and was at least partly determined by the relationship between a unit's preferred direction and the directions of stimulus motion. Across the population, neuronal sensitivity was significantly lower than psychophysical sensitivity, although the most sensitive units matched or exceeded behavioral sensitivity to the segmentation cues. Furthermore, there was a significant trial-by-trial co-variation between firing rates and perception, indicating that MT signals play a role in the segmentation process. Units with preferred directions that optimized their sensitivity to differences in plaid segmentation cues, and that tended to signal global motion in stimuli with multiple local directions, displayed the largest perceptual correlations. Here, we address a few potential concerns before situating these results with respect to previous work.

A primary concern in studies employing bi-stable stimuli in animal models is that behavioral responses might not be based on the dimension of interest. For example, our monkeys could have been reporting their perception of texture direction independently of their perception of plaid coherence. Two aspects of the data suggest this was not the case. First, consistent with a previous report[10], varying the relative angle separating plaid component directions systematically altered the probability of a coherent percept. Second, on average, this effect was similar for patterns containing and lacking a texture cue. Together, these observations suggest monkeys' answers consistently reflect their perception of coherence/transparency.

Another potential concern is that we did not optimize the parameters of plaid motion on a case-by-case basis. In much prior work comparing neuronal and psychophysical sensitivity the stimulus is individually tailored for each recorded unit[31,32,34,36–45]. Here, we used the same two directions of plaid pattern motion, regardless of each unit's direction tuning. This design allowed us to examine how sensitivity varied as a function of the overlap between plaid motion and preferred direction, however, it did not provide an a priori basis for determining whether a cell preferred coherent or transparent plaids. Thus, we relied on empirical criterion, using each unit's responses to textured plaids to assign preferred and null tags to each category of plaid motion. It is possible, though unlikely, this could have systematically biased the results of our sensitivity and CP signal detection analyses, potentially inflating either measure. However, several facets of both the analysis and the data, addressed in turn below, suggest this was not the case.

First, assigning preferred (null) designations to stimuli that elicited greater (lesser) activity does not influence the discriminability of these response distributions. Rather, it only ensures that neurometric and psychometric functions have identical sign and thus can be directly compared. Second, responses that were used to calculate CP ("error" trials for textured plaids and all trials for plaids without texture contrast) were not included in the regression analysis that determined whether each cell "preferred" coherent or transparent motion. This ensured that choice effects did not bias the preferred/null designation and in turn yield significant choice probabilities.

Studies by Newsome and colleagues[36,39,46,47] first established a role for MT in coarse judgments of motion direction. Subsequent reports have marshaled evidence for MT's involvement in depth[34,44,48–51] and speed[32,52] judgments, in fine-direction discrimination[33] and in the perception of three-dimensional structure-from-motion[31,53,54] (3-D SFM). We have extended these results in two important ways. First, we provide evidence that MT responses contribute to the perceptual segmentation of visual motion signals. Second, we observed a relationship between MT pattern direction selectivity and this choice signal.

Conceptually, the present results are most similar to the work on 3-D SFM, as both constitute complex bi-stable percepts involving motion and depth ordering. Dodd et al.[31] found a large choice probability (0.67) in a task in which monkeys reported the direction of rotation of a bi-stable 3-D SFM cylinder. We found a much smaller choice effect for bi-stable plaid stimuli (on the order of 0.55 across both monkeys). Because estimates of CP depend upon the choice ratio[30,31], it can be difficult to interpret CPs obtained under different conditions in different tasks. However, the magnitude of the choice effect we observed was consistent across zero- and low-texture contrast plaids as well as when we pooled across low/zero texture contrast stimuli to increase power. Thus, this difference in CP is unlikely to be due to differences in choice ratios across the datasets.

When contrasted with the strong and qualitatively distinct perceptual states generated by both 3-D SFM stimuli and bi-stable plaid patterns, the modest changes in MT firing rates that accompany perception in the latter case seem perplexing. One possibility is that we underestimated the choice effect by calculating firing rates over the entire stimulus period. In contrast to the 3-D SFM case[31], in which differences in MT activity developed around 250 msec into a trial and then rose steadily throughout, our analysis of the temporal dynamics of the choice signal (cf. Fig. 6i) revealed a near chance CP until almost 500 msec after stimulus onset for both monkeys. Furthermore, following a sharp rise around this time, we observed a fluctuation in CP throughout the remainder of the trial. Hupe and Rubin[55] reported frequent alternations in human observers' perception of bi-stable square wave plaids during extended trials. Although our stimuli were only presented for 1.5 s, it is possible our monkeys' perception also varied between coherence and transparency during a trial (with their

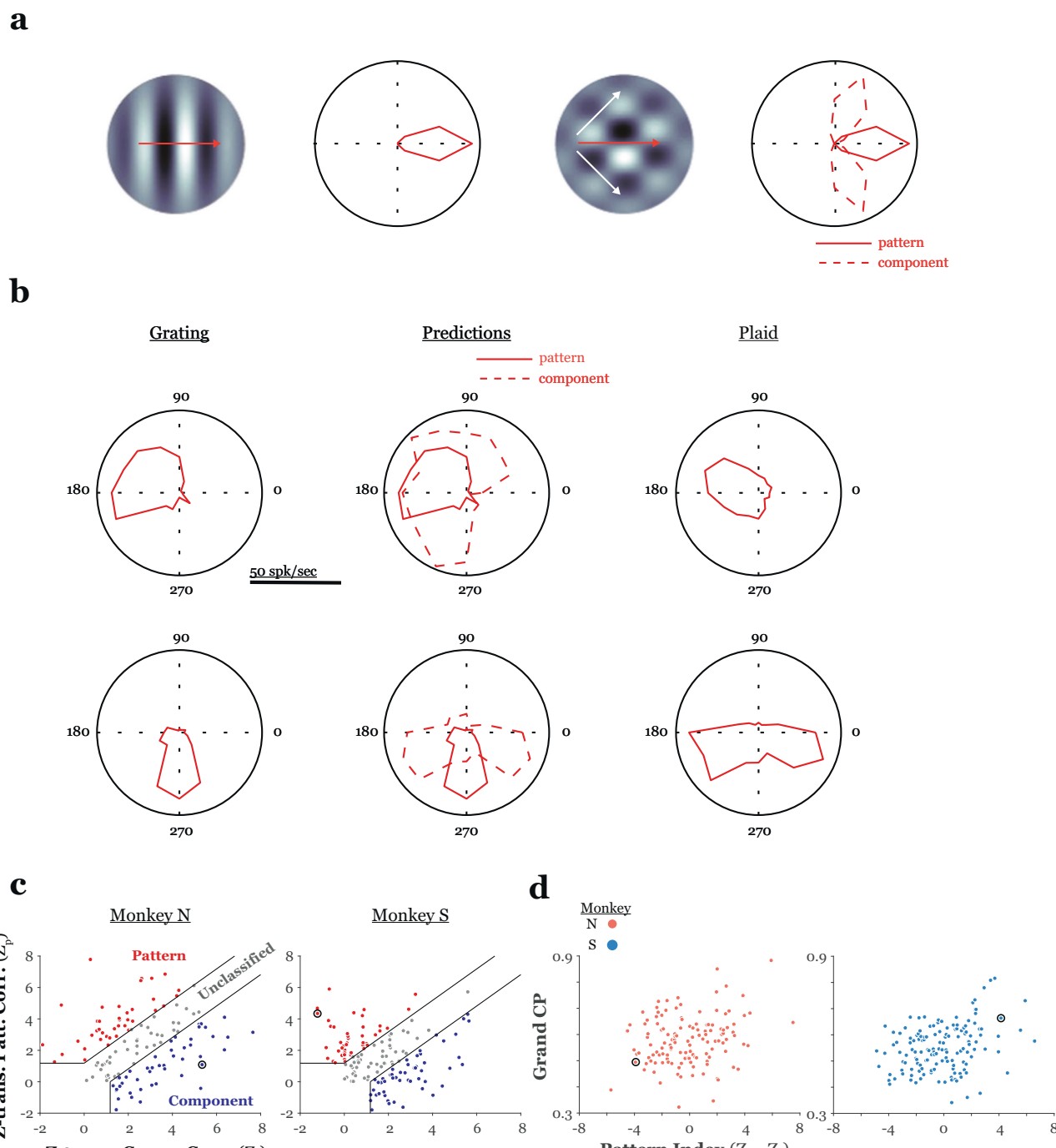

**Fig. 7 | Relationship between MT pattern-motion sensitivity and choice-related activity in the plaid segmentation task. a** Schematic illustration of pattern-component tuning stimuli and hypothetical grating (left) and plaid direction tuning curves (right) (see Materials and Methods). Briefly, if a cell integrates across plaid components to signal pattern motion, then one would expect tuning curves to be identical for single grating and plaid stimuli (last column, solid curve). Conversely, if a cell did not integrate component directions to signal pattern motion, one would expect a bi-lobed tuning curve, with a peak for each direction of plaid motion that translates a single component in the unit's preferred direction (final column, dashed curve). **b** (left) Sine grating direction tuning curves for the unit shown in Figs. 3 and 4 (top row – unit from Figs. 3a, b and 4a, b (top); bottom panel – unit from Figs. 3c, d and 4a, b (bottom)). (middle) Pattern and component predictions calculated from the grating tuning profiles. (right) Plaid tuning for these units. The unit in the top (bottom) panel was classified as a pattern (component) cell. Note that there is not a one-to-one correspondence between pattern-component classification and a cell's coherent/transparent motion preference (cf. textured plaid responses for these units in Fig. 4a). **c** The z-scored pattern partial correlation coefficient (ordinate) is plotted versus the z-scored component partial correlation coefficient (abscissa) for all cells recorded from monkey N (left) and S (right). Thick lines represent significance criteria used to classify cells. **d** Grand choice probability (ordinate) is plotted versus pattern index ($Z_p$ – $Z_c$) (abscissa). Data in the left (right) panel are from monkey N (S). Black circles highlight data from the example units. In both animals, there was a significant correlation between grand choice probability and pattern index, indicating a greater perceptual correlation for cells that signal pattern direction in stimuli with multiple component directions.

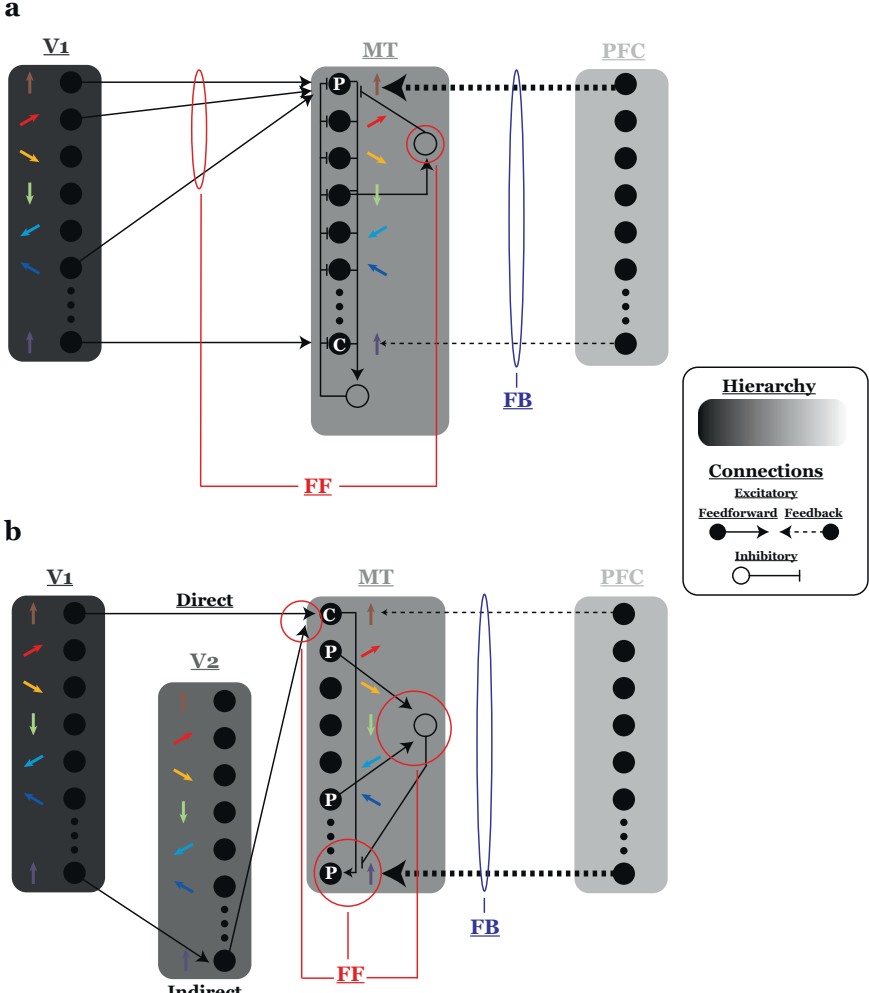

**Fig. 8 | Possible circuitry underlying the relationship between pattern selectivity and choice probability. a** A two-stage model of component- and pattern-direction selectivity and the potential influence of top-down feedback on choice-related activity in MT. Here, pattern direction selectivity (PDS - "P") at the MT stage arises through: (i) broad sampling over direction selective inputs consistent with a particular pattern velocity, and (ii) strong tuned suppression. MT stage component direction selective (CDS) cells ("C") sample narrowly over input directions and lack strong tuned suppression. Untuned suppression confers gain control in both populations. Colored arrows represent units' preferred direction. Only a subset of V1-MT connections and a single pattern- and component-direction selective unit are illustrated for clarity. In the case of a feedforward (FF) interpretation of our results, the broader input tuning and strong tuned suppression in PDS cells (highlighted in red) generates a larger difference in activity in response to patterns with multiple motions. This population drives upstream decision circuits and biases perception in our segmentation task. Conversely, in the feedback (FB) case,

perceptual decisions are generated in upstream circuits by both sensory evidence and cognitive biases and a greater influence of top-down FB on PDS cells (thicker line) generates the choice signal. **b** Illustration of an alternative model of CDS and PDS units. Here, PDS signals in MT arise via not only direct V1 input but also via indirect inputs from a V1-V2-MT pathway. The model indirect pathway is configured to confer selectivity for texture boundaries (plaid overlap regions). MT stage CDS cells perform a weighted sum on direct and indirect inputs and send outputs to PDS units. PDS tuning arises via competitive inhibition. Again, only those connections necessary to sketch the basic model architecture are shown. Here, distinct FF mechanisms from those posited in **a** could be responsible for driving greater variability in PDS cell plaid responses, which would, again, drive biases in decision circuits. Alternatively, greater CP in PDS cells could still be the result of a bias in the strength or efficacy of FB connections to PDS cells. There is evidence supporting both two- and three-stage models of MT PDS and FF and FB explanations for CP.

answer reflecting their final perception at the time of the choice cue). Therefore, a reaction-time version of our task, or a design in which monkeys were able to continuously report their perception, would be expected to yield a much larger choice effect. A final possibility is that MT signals are readout in a different manner across the two tasks. Although it has long been suggested the CP signal arises from sensory decoding and correlated noise[56], Gu and co-workers[57] found that varying pooling strategies, rather than the level of correlated variability, in a computational model better explained variations in CP across dorsal medial superior temporal (MSTd) neurons in a heading discrimination task. It is possible the smaller choice effect we observed in MT reflects broad pooling across many weakly informative neurons to generate a perception of coherence or transparency. In any case, the

independent demonstrations that MT responses are significantly correlated with perceptual judgments both when local motion signals must be grouped into one or two objects (bi-stable plaids), or into separate surfaces of a common object (3-D SFM) strongly suggests MT responses play a role in using visual motion information to segment complex images into multi-object scenes.

As mentioned above, ours is the first report of a link between MT pattern cell activity and perception. As formulated in the original two-stage model of Movshon and colleagues[21], pattern cells represent the MT output stage. However, more recent work suggests that pattern and component cells represent distinct ends of a continuum, with parametric differences in receptive field structure responsible for the spectrum of pattern-component tuning[16,20,26,58–61]. Thus, our

finding of a significant correlation between CP and PI is similar to the relationship between the symmetry of binocular disparity tuning and CP in a depth discrimination task[34,41,42], or between direction tuning profiles and CP in a fine direction discrimination task[33]. Wang and Movshon[62] analyzed a large pool of MT direction selective cells and found that, on average, pattern index was correlated with numerous tuning properties, suggesting that pattern selectivity is present with many other types of signals that can be readout from the MT population. Thus, it will be important for future studies of the link between MT activity and subjective perception to determine whether pattern index is similarly correlated with the choice signal for other tasks and stimuli or whether this relationship is special to the case of perceptual segmentation.

In a similar vein, Nienborg and Cumming[42] found that although near and far tuned binocular disparity selective cells in V2 were equally sensitive in a depth discrimination task, only the population of near preferring cells exhibited significant CPs. However, re-training a monkey to preferentially weight far disparities resulted in significant CPs in far preferring cells. Other studies have also reported a training history dependence of either a perceptual correlation[34,40,63] or a causal link[48] between MT activity and disparity discrimination. It is possible the relationship between CP and pattern direction selectivity that we observed might reflect the particular strategy monkeys used to solve our task, rather than a special role for pattern selective signals in visual motion perception. It will be important to determine in future work whether training history has a significant influence in determining which MT signals are preferentially and flexibly weighted to generate segmentation judgments.

Stoner and colleagues[14,23] first reported that changing the luminance of plaid overlap regions predictably affected human observers' coherence and transparency reports as well as the direction tuning of macaque MT neurons. These authors found that when the luminance of overlap regions was physically consistent with transparency, observers' reported more transparent percepts and MT neurons' signaled the component grating motions. Conversely, when overlap luminance was physically inconsistent with transparent overlay, observers perceived coherent motion and MT neurons signaled the global motion of the pattern. Thus, these studies showed that physical changes in the visual stimulus that reliably affect segmentation reports also induce predictable changes in MT firing. More recent work has followed in this vein, examining what signals in MT track the perceptual appearance of complex stimuli[18,24,64]. For example, it has been shown that a subset of MT neurons exhibits bimodal tuning to random dot kinematograms (RDKs) that have two directions that are separated by less than the cells' tuning bandwidth for unidirectional RDKs[19,25]. Observers always perceive the former patterns as transparent motion, even though most MT neurons show unimodal tuning in response to these stimuli and a simple averaging across all MT units yields a unimodal population response. Thus, the subpopulation of cells showing bimodal tuning could form the neural substrate for this perception. Intriguingly, in the marmoset this population was shown to correspond to PDS cells when tested with conventional grating and plaid stimuli[19].

Our results go a step further than the above in a manner essential to establishing MT's role in perceptual segmentation. At its core, segmentation is a subjective phenomenon; numerous multi-stable visual displays illustrate the ability of the visual system to organize and interpret a constant stimulus in more than one way[2]. The simultaneous collection of neural response and perceptual report in our study allowed us to examine the co-variation between MT firing rates and the perceptual interpretation of a constant stimulus. Having shown such a link, we recognize that the direction of causality has yet to be determined; that is, further experiments will be necessary to determine whether the perceptual segmentation signal we observed is, as some have claimed[65-67], a bottom-up process or a top-down signal fed back

to sensory cortex from higher areas[68-70] (Fig. 8). The report of an even larger fraction of pattern selective cells in MSTd[71] – one of MT's principle cortical targets – suggests that extending these experiments to include simultaneous recordings from MT and MSTd will be a good first step in furthering our understanding of the neural mechanisms of perceptual segmentation.

## Methods

### Subjects and surgery
Two adult macaque monkeys (*macaca mulatta*), one male and one female (7 and 5 years old, respectively), weighing between 4.5 and 9.0 kg were used as subjects in this study. Prior to all experiments, in aseptic surgeries, animals were implanted with a custom-made recording chamber for vertical electrode approach to area MT, a stainless steel post for head restraint (Crist Instruments, Hagerstown, MD), and a scleral search coil for measuring eye position (Cooner Wire, San Diego, CA). All protocols conformed to United States Department of Agriculture (USDA) regulations as well as the National Institutes of Health (NIH) guidelines for the humane care and use of laboratory animals, and were approved by the University of Chicago Institutional Animal Care and Use Committee (IACUC).

### Visual stimulus generation
All visual stimuli were presented within a circular aperture on a black or gray background. During recording sessions, the location and diameter of this aperture were adjusted to match the classical receptive fields of the neurons at the electrode tip. We used two broad classes of visual stimuli: psychometric stimuli and tuning stimuli.

**Psychometric stimuli.** Psychometric stimuli were plaid patterns created by superimposing two square-wave gratings that drifted in a direction perpendicular to their orientation (20 cd/m², 50% contrast, 50% duty cycle, 5 degs/sec) (Fig. 1b). It has previously been shown that human observers report these plaid patterns as bi-stable stimuli, sometimes appearing as a single pattern moving in one direction (coherent motion) other times as two separate surfaces moving in different directions (transparent motion)[11]. The component gratings that comprised the plaid patterns were oriented symmetrically – at an inter-grating angle of between 95° and 130° (drawn from the set: 95°, 100°, 105°, 115°, 120°, 125°, 130°, no neuronal isolations were maintained throughout the sessions with a 115° inter-grating angle but we include the psychophysical data here) – around either 90° or 270° (pattern directions). Only one inter-grating angle was used in each session; during each session, pattern direction was randomly selected on each trial from the two possibilities.

To disambiguate plaid perception and provide an empirical basis for operant reward we introduced random dot texturing to the light bar phase of each plaid component[72]. This was accomplished by increasing or decreasing – by a fixed amount – the luminance of a randomly selected subset of pixels (Fig. 1c). The direction(s) of texture motion provided a powerful cue that biased observers' perception towards either coherent or transparent motion (Fig. 1c). In the coherent condition, all texture, regardless of which component grating the texture overlay, was translated in the pattern direction (Fig. 1c, coherent). In the transparent condition, texture was translated normal to the orientation of the grating it overlaid (Fig. 1c, transparent) (Supplementary Movie 1). To manipulate task difficulty, in the majority of sessions, across trials the Michelson contrast (Lmax-Lmin/Lmax+Lmin) of this texture cue was drawn from the set (−80, −40, −20, −10, −5, 0, 5, 10, 20, 40, 80). Contrast was defined relative grating luminance (so a value of 80% contrast would result in texture of 36 or 6 cd/m²). In 6 sessions from monkey N and 5 from monkey S we used a tighter range of texture contrasts (−30, −20, −15, −10, −5, 0, 5, 10, 15, 20, 30) for which psychophysical performance followed the same pattern as with the full range of contrasts but did not saturate.

**Tuning stimuli**. Tuning stimuli were sinusoidal gratings (50% contrast, 1 cycle/deg, 5 deg/sec) moving in one of 16 equally spaced directions, or sinusoidal plaids (formed by superimposing two sinusoidal gratings with a relative angle of 135°) moving in these same pattern directions.

## Tasks and training

During all experiments, monkeys sat in a primate chair positioned 57 cm from a 21-inch CRT monitor subtending 40 by 30 degrees of visual angle (ViewSonic PF815, 800 × 600 resolution, 60 Hz refresh rate). Task timing and visual stimulus presentation were under the control of networked computers running custom software for the real time control of behavioral neurophysiology experiments[73]. Monkeys were tested in two tasks: a psychometric task, used to assess their perception of coherence/transparency in our psychometric stimuli, and a tuning task, used to assess some of the basic visual response properties of single units in area MT.

In the psychometric task, monkeys were trained to report their perception of plaid motion (coherent or transparent). During psychometric trials, monkeys were required to maintain fixation within 0.5 degrees of a central fixation spot while a plaid stimulus was presented for 1.5 s to an eccentric portion of the visual field (between 3 and 8 degrees from fixation). Following stimulus offset, choice targets were presented 5 degrees directly above (coherent target) and below (transparent target) the central fixation point. Animals had up to 1 s to indicate, via a saccade to the appropriate target, whether the preceding pattern had appeared as coherent or transparent motion. Correct answers were rewarded with auditory feedback and a bolus of water, incorrect answers and trials on which the monkey prematurely broke fixation were followed by a brief time-out period. When plaid stimuli lacked a texture cue (i.e., texture contrast = 0%), and thus a physical basis for determining a correct response, animals were rewarded randomly (50/50 odds). During initial training, only patterns containing a high contrast texture cue were presented, with one texture contrast per training session. As performance for a given texture contrast saturated, the texture contrast was reduced. This process continued until monkeys' achieved stable performance at all contrasts; at this point in training monkeys were then exposed to sessions in which all texture contrasts were interleaved in a pseudorandom (blockwise) fashion. During recording sessions, texture contrast was selected pseudorandomly on each trial.

In tuning trials, monkeys were only required to maintain fixation on a central fixation point while stimuli were presented at a location in the peripheral visual field. Tuning stimuli were presented for 1 s; monkeys were given auditory feedback and a bolus of water for maintaining fixation within 0.5 degrees of the fixation point for the duration of stimulus presentation. Trials in which the monkey broke fixation before the disappearance of the stimulus were followed by a brief time-out period.

## Electrophysiological recordings

Neural activity was recorded using bundles of 1–5 epoxylite-coated tungsten microelectrodes (FHC electronics, Bowdoinham ME; impedance 0.5–1.5 MΩ at 1 kHz) inserted into cortex within stainless steel guide tubes held in place by a custom made plastic grid secured in the recording chamber[74]. Electrodes were tucked inside the guide tube during penetration of the dura mater and then advanced into MT by means of a hydraulic stepping motor (FHC electronics, Bowdoinham ME). MT neurons were identified based on physiological criteria and the stereotyped pattern of gray and white matter transitions encountered during the electrode penetration. Neural activity was amplified and bandpass filtered (300–8 kHz), voltage deflections that exceeded a user-determined threshold were timestamped and stored to disk.

## Neuronal sample selection

We isolated signals from 314 single-units in 86 separate recording sessions (36 from monkey N and 50 from monkey S – cf. Supp. Figure 1b for number of sessions at each inter-grating angle). RF centers ranged from 3 to 8 degrees eccentricity (the segmentation task became difficult for the monkeys to perform beyond this range), RF diameter roughly scaled with eccentricity. During each experiment, neural activity was recorded while monkeys performed our segmentation and fixation tasks. Only well isolated MT units from which we were able to record full plaid segmentation and pattern-component tuning datasets, and for which performance in the segmentation task met our criterion (see Results) were included in the final sample (120 units from monkey N and 157 units from monkey S – cf. Supplementary Fig. 2b for sample sizes at each plaid inter-grating angle). In cases in which we applied further exclusion criteria (e.g., the CP analysis); sample sizes are given in each plot.

We used responses to tuning stimuli to characterize the direction tuning of MT neurons and to classify neurons as pattern- or component-direction selective[21]. Direction tuning curves were estimated by sorting trials according to the direction of grating or plaid motion then averaging firing rates during the 1-second stimulus presentation. Background rates were computed from epochs immediately preceding and following trials when no stimulus was present. Preferred direction and tuning bandwidths (full width at half maximum – FWHM) were extracted from Von Mises functions fit to each unit's direction tuning curve[33].

In addition to estimates of preferred stimulus direction, responses to tuning stimuli were used to classify units as pattern- or component-direction selective[21]. If a unit responded primarily to the local motion of a pattern's oriented components, then we would expect its responses to a plaid to be the sum of its responses to each component grating (component prediction). Conversely, if a cell were to encode the global two-dimensional motion of a plaid pattern, we would expect plaid responses to be unaffected by component orientation; i.e., direction tuning curves measured with gratings and plaids should be identical (pattern prediction). We used partial correlation analysis to determine the extent to which MT responses correlated with both the pattern and component predictions[21]. To generate a continuous index of pattern-component selectivity (pattern index – PI) and test whether a cell's responses were significantly correlated with either the pattern or component predictions we transformed partial correlation coefficients to z-scores using Fisher's R-to-z transformation[35]. To facilitate comparisons with extant datasets[35,75], we used a significance criterion of $p = 0.1$ – corresponding to a z-score of 1.28 – for pattern-component classification purposes. That is, classification as a pattern cell required a cell's z-scored pattern partial correlation coefficient ($Z_p$) to exceed its z-scored component partial correlation coefficient (or 0 if $Z_c$ was negative) by 1.28. Component classification proceeded by the same method, with $Z_c$ needing to exceed $Z_p$ (or 0 if $Z_p$ was negative) by 1.28. Units that met neither the pattern nor the component criteria were termed unclassified.

## Analysis of perceptual reports

A psychometric function relating performance on our segmentation task to the magnitude (texture contrast) and sign (transparent/coherent) of segmentation cues in our plaid stimuli was generated for each session by calculating the probability of a coherent judgment for each texture contrast (where, by definition, transparent patterns assume negative values and coherent patterns positive values). These data were fit to Gaussian cumulative distribution functions using maximum likelihood methods. Estimates of the point of subjective equality (PSE – texture contrast at which coherent/transparent judgments split 50/50) and threshold were extracted from this function as the mean and mean plus one standard deviation, respectively.

## Comparing neuronal and psychophysical responses

We used two separate methods to quantify the relationship between neural activity, the physical configuration of our psychometric plaid stimuli and monkeys' perceptual reports.

To directly compare neuronal to psychophysical sensitivity and to quantify trial-by-trial variations in neural activity with variations in the perception of a constant stimulus, we used a receiver operating characteristic analysis (ROC) to calculate each cell's neurometric function and choice probability (CP), respectively. These methods have been described extensively elsewhere[30,36]; thus, we limit our description here to a few key points.

To first assign a coherent/transparent motion 'preference' to each neuron we regressed average firing rates during psychometric trials on texture contrast. We used the sign of the slope coefficient to classify a neuron as preferring coherent (positive) or transparent (negative) motion. For each cell, this regression was done separately for each direction of pattern motion and was limited to stimuli with non-zero texture contrast. The analysis then proceeded separately to calculate sensitivity and choice probability.

For the sensitivity analysis, the distribution of responses to preferred (e.g., coherent texture cue) and non-preferred (e.g., transparent texture cue) stimuli with a particular magnitude of texture contrast (e.g., 10%) were used to calculate a ROC curve. It has been shown that the area under this curve corresponds to the probability that an ideal observer (that is, a noise free readout mechanism), given only a random draw from one of these two distribution of firing rates, will correctly identify the stimulus presented on a given trial[76]. Performing this analysis separately for each texture contrast yields a neurometric function that can then be subjected to the same analysis described above for psychophysical data. Only responses to patterns that contained non-zero texture contrast, and for which the monkeys answered correctly, were included in the sensitivity analysis.

In contrast to the procedure outlined above, for the calculation of choice probability, trials were sorted according to the magnitude (e.g., texture contrast) and sign (e.g., coherent) of the texture cue. Responses were then binned according to whether the animal reported the stimulus on a given trial as coherent or transparent motion. ROC curves were calculated from these two distributions. The area under this curve reflects the probability that an ideal observer, given only a random draw from these distributions, will correctly predict the monkeys reported perception on a given trial[30].

### Reporting summary

Further information on research design is available in the Nature Research Reporting Summary linked to this article.

## Data availability

These data are currently being used to inform and validate an on-going computational modeling study; thus, they are not currently publicly available at this time but we are prepared to make them available upon reasonable request to the corresponding author. Source data are provided with this paper.

## Code availability

No novel or non-standard methodology was used in the data analyses reported in this manuscript. All functions used for analysis were standard implementations in MATLAB 2017a, Mathworks (Natick, MA). All custom code used for data analysis is publicly available at: https://github.com/clarkam80/mtperceptualsegmentation-dataanalysis.

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

## Acknowledgements
We wish to thank Drs. P. Hosseini, B.B. Scott, M. Mascaro, and S. Santhakumar for expert technical assistance and advice during all phases of the project. We would also like to thank Drs. A. Angelucci and L. Nurminen for insightful discussion of earlier drafts. Support for this work was provided by EY13138 (D.C.B.) from the NEI.

## Author contributions
A.M.C. and D.C.B. conceived and designed this study, A.M.C. performed the experiments and collected the data, A.M.C. and D.C.B. analyzed the data, and A.M.C. and D.C.B. wrote the paper

## Competing interests
The authors declare no competing interests.
