## [Peer Review File · Nature Communications]

A Neural Correlate of Perceptual Segmentation in Macaque Middle Temporal Cortical AreaREVIEWER COMMENTS

Reviewer #1 (Remarks to the Author):

This paper reports the finding that neuronal responses in the middle-temporal (MT) cortex of macaque monkeys are correlated with the animals' choices of perceptual coherence and transparency. Previous studies have found that neurons in area MT show choice-related activity when monkeys perform direction or disparity discrimination tasks, or a structure-from-motion task. Other studies have shown that MT direction tuning changes with visual stimuli to reflect perceptual segmentation or integration. However, previous studies have not shown MT activity reflects the choice of the animal on perceptual coherence/transparency. The finding of this paper fills a critical gap in the literature and provides an important piece of evidence supporting the idea that area MT plays a role in perceptual integration and segmentation.

A challenge in this type of study is to ensure that animals indeed report the subjective percept of the visual stimuli, rather than relying on other stimulus cues to perform the task. The authors took considerable care to design the behavioral task. They took advantage of the stimuli that van den Berg and van de Grind (1993) designed by adding random-dot texture moving either in the pattern or component directions on plaid stimuli so the perceived motion directions were disambiguated. The animals' choices for coherence and transparency changed systematically with the contrast of the random-dot texture. A critical condition is when the texture contrast was set to zero. In this case, no disambiguating texture was added. The authors found that with zero and low contrast random-dot texture, MT neurons showed choice-related activity and had a mean choice probability (CP) significantly greater than 0.5. The authors further showed that CP and the pattern-index of the neurons are positively correlated. In other words, neurons that are more selective to pattern directions, characterized by plaid stimuli without texture dots, are likely to show a larger CP when animals reporting coherence vs. transparency. These are interesting results. The experimental design is solid and the paper is generally well written.

Major:

1. In Figure 3, the authors showed the results from a representative neuron in response to two types of stimulus cue (coherent or transparent) at different levels of texture contrast. What was not shown is how the response from a representative neuron changes with the animal's choice at different levels of cue contrast, and critically, at zero contrast. This can probably be introduced later in the paper, after the current Figure 4 and before Figure 5. For each (identical) stimulus condition, showing PSTHs for two different choices in the same plot would be helpful to see the effect of choice.

2. The authors used different inter-grating angles between 95° and 130° and showed in Figure 2d that the Point of Subjective Equality (PSE) increases with the inter-grating angles. From the description on page 20, line 590 to 594, it seems that each neuron was only tested with one inter-grating angle. What were the exact values of the inter-grating angles used? How many neurons were tested at each inter-grating angle? Is there a systematic difference among results obtained with different inter-grating angles? For neurons that had preferred direction aligned (or close to) the direction of one of the component directions or the pattern direction, a larger inter-grating angle, and therefore a larger angle difference between the pattern and component directions would likely result in a larger CP. Is this the case? More data shall be presented.

3. On page 5, lines 136-137, it is stated that "monkey's responses on zero text contrast were significantly affected by changes in inter-grating angle". This is an important validation. Is this result shown anywhere in the paper?

4. The authors described the dependency of neuronal sensitivity on neuron's preferred direction relative to the stimulus direction(s) (lines 190-209 on page 7). I suggest to include a data figure showing neuronal sensitivity as a function of the relationship between the preferred direction and the stimulus direction(s).

5. Following point #3, the authors showed the relationship between CP and neuronal sensitivity (Fig. 5c, f), but did not show directly the relationship between CP and relative direction preference, which could be informative.

6. The relationship between CP and the pattern index shown in Figure 6d is fairly clear. However, a potential confounding factor is the neuron's preferred direction relative to the stimulus direction(s). One possibility is that perhaps the pattern cells in the data sample happen to have their preferred directions better aligned with the pattern direction of the stimulus. Does the trend (CP is positively correlated with the pattern index) still hold if one looks at the neuronal responses to upward direction and downward direction separately (using partial correlation to control for sensitivity)?

7. Furthermore, does the trend between CP and pattern index still hold if only the CP from zero texture contrast is used?

8. Are the regression lines shown in Figure 5c, f, and Figure 6d from type I or II regression? Type II regression should be used if not already.

Minor:

1. For the two example neurons shown in Figure 6, indicate their pattern indices and CPs or mark the corresponding data points in the scatter plots of Figure 6c and d.

2. It would be helpful for readers to see the visual stimuli used in this study. Perhaps include a link to the movies.

3. The papers by McDonald, Clifford, Solomon, Chen, Solomon (2014) in the Journal of Neurophysiology and Xiao and Huang (2015) in the Journal of Neuroscience both looked at the relationship between MT neurons' direction tuning properties relevant to motion segmentation and pattern/component selectivity to plaid stimuli. It may be useful to compare and contrast the current finding with these previous results.

4. How many neurons (units) were recorded and included in the data analysis? How many from monkeys N and S, respectively?

5. What is the range of the stimulus diameter?

6. What is the range of the RF eccentricity?

7. What are the luminance values for L_{max} and L_{min} at 80% and 0% Michelson contrast?

Reviewer #2 (Remarks to the Author):

This paper aims to show that Area MT in monkeys has a direct correlation with the perception of motion. Several authors have already demonstrated that MT appears to be involved in 'driving' the perception of motion. The specific issue investigated here is whether MT (and particularly one cell type in MT) is involved in the perception of motion segmentation judgements. The most important finding is that a subset of cells in Area MT, which have been labelled pattern cells based on a single test, is particularly correlated with segmentation.

The work is certainly of interest to people in the field. However, as the authors mention themselves, one of the largest problems in my mind is the importance of the finding in a broader context. Pattern and Component cells are defined based on a very restricted stimulus-type: namely plaid patterns. Pattern and component cells do not form two distinct groups but, rather are defined based on a

statistical test that splits cells into three groups that do not naturally split into three separate clusters. There is no border between these groups (i.e. statistics don't allow you to identify three groups with dips between them). In truth, there is a continuum and most cells are a bit of both extremes. Plaid patterns contain restricted amounts of information on motion direction, so the lack of clear boundaries is not surprising. Therefore, saying that pattern cells are more closely linked to the perception of transparent motion appears philosophically to be a rather small increment. For example, what would happen if more certainty were built into the stimulus, such as occurs immediately, perceptually, when terminators appear in the RF? Would cells defined as pattern-selective differ from the other MT cells in this case and would this influence the bias of the single-cell responses? Would the 'special' case for pattern cells persist?

The work was conducted in behaving primates and this is a very challenging and time consuming technique. To the best of my knowledge the work was conducted at the highest level and the statistics are appropriate. I believe that the experiments could be repeated by following the methods. I am not in a position to analyse the data to see if I can reproduce the same result.

Therefore, while the work is of the highest quality, the paper suffers in my view from not clearly showing how the MT population code as a whole deals with transparency and segmentation judgements. I feel that a model that incorporates all of the response types in MT would be essential in trying to answer this. The authors could develop a data-driven model that uses their hard-won monkey data. As it stands, I feel that the paper attempts to highlight a correlation between perception and just one type of MT cell (i.e. pattern cells).

Reviewer #3 (Remarks to the Author):

This has been one of the easiest papers I have reviewed. The authors have done an excellent job addressing an open question about the role of area MT in discriminating between transparent versus coherent motion. Results show that MT responses signal transparent versus coherent motion, are correlated with behavior, and are stronger in pattern direction cells, the cells that were previously hypothesized to play a more direct role in motion perception.

The experiment is well designed and potential shortcomings are properly discussed. The data analysis follows classic work, is thorough, and acknowledges open questions. Finally, the conclusions are consistent with the observations, and the potential caveats are properly discussed.

I only have one comment that I would like the authors to discuss. When looking at transparent versus coherent motion stimuli, one gets the feeling that the two are very different perceptually; intuitively, the effect doesn't seem to boil down to a few spikes more or less in one direction or another. Do the authors really believe that a just-above-chance CP is all it takes to flip the perception qualitatively? I know this type of thinking is not common in classical work in area MT, but I think it is useful for the readers to hear the authors' speculation on how such discrete precepts might emerge from small graded differences in firing rates.

Otherwise, a nice piece of work!

-Mehrdad Jazayeri

We would like to thank Dr. Jazayeri and the two anonymous reviewers for their timely and constructive criticism. We appreciate the enthusiasm the reviewers' expressed for the findings and are immensely grateful for their time and effort in helping to improve the rigor and quality of the work. We have now revised both the manuscript and figures according to the reviewers detailed comments. To facilitate review of the revised version all amended text discussed below has been highlighted in red in the revised manuscript. Additionally, we have highlighted the section, page, and line numbers of all key revised portions of text in the responses to relevant reviewer comments. Below, please find these point-by-point replies to the reviewer's concerns.

Reviewer #1

~~This paper reports~~ the finding that neuronal responses in the middle-temporal (MT) cortex of macaque monkeys are correlated with the animals' choices of perceptual coherence and transparency. Previous studies have found that neurons in area MT show choice-related activity when monkeys perform direction or disparity discrimination tasks, or a structure-from-motion task. Other studies have shown that MT direction tuning changes with visual stimuli to reflect perceptual segmentation or integration. However, previous studies have not shown MT activity reflects the choice of the animal on perceptual coherence/transparency. The finding of this paper fills a critical gap in the literature and provides an important piece of evidence supporting the idea that area MT plays a role in perceptual integration and segmentation.

A challenge in this type of study is to ensure that animals indeed report the subjective percept of the visual stimuli, rather than relying on other stimulus cues to perform the task. The authors took considerable care to design the behavioral task. They took advantage of the stimuli that van den Berg and van de Grind (1993) designed by adding random-dot texture moving either in the pattern or component directions on plaid stimuli so the perceived motion directions were disambiguated. The animals' choices for coherence and transparency changed systematically with the contrast of the random-dot texture. A critical condition is when the texture contrast was set to zero. In this case, no disambiguating texture was added. The authors found that with zero and low contrast random-dot texture, MT neurons showed choice-related activity and had a mean choice probability (CP) significantly greater than 0.5. The authors further showed that CP and the pattern-index of the neurons are positively correlated. In other words, neurons that are more selective to pattern directions, characterized by plaid stimuli without texture dots, are likely to show a larger CP when animals reporting coherence vs. transparency. These are interesting results. The experimental design is solid and the paper is generally well written.

We thank the reviewer for their appreciation for the difficulties inherent in conducting this type of study and are encouraged by their positive predisposition as to the reliability and potential impact of our findings.

Major Comments

1. In Figure 3, the authors showed the results from a representative neuron in response to two types of stimulus cue (coherent or transparent) at different levels of texture contrast. What was not shown is how the response from a representative neuron changes with the animal's choice at different levels of cue contrast, and critically, at zero contrast. This can probably be introduced later in the paper, after the current Figure 4 and before Figure 5. For each (identical) stimulus condition, showing PSTHs for two different choices in the same plot would be helpful to see the effect of choice.

We thank the reviewer for highlighting the value to the reader in presenting specific examples of how individual MT neurons' responses are affected by perceptual choice. Accordingly, we have added plots in the style the reviewer suggests above (separate PSTHs for each neuron on a single axis, with trials sorted by perceptual choice) to the new Supplementary Figure 3. Additionally, based on this comment and others from the reviewer below, we have revised the overall presentation of data from example MT cells so that responses from 2 units to textured plaids now appear in the new Figure 3. Furthermore, again in response to comments from this Reviewer, we have highlighted individual data points from these examples in the summary plots in the scatterplots in the new Figures 5, 6, and 7. We believe this both better illustrates the range of responses in the dataset and better helps the reader visualize a given effect in different regions of summary plots.

2. The authors used different inter-grating angles between 95° and 130° and showed in Figure 2d that the Point of Subjective Equality (PSE) increases with the inter-grating angles. From the description on page 20, line 590 to 594, it seems that each neuron was only tested with one inter-grating angle. What were the exact values of the inter-grating angles used? How many neurons were tested at each inter-grating angle? Is there a systematic difference among results obtained with different inter-grating angles? For neurons that had preferred direction aligned (or close to) the direction of one of the component directions or the pattern direction, a larger inter-grating angle, and therefore a larger angle difference between the pattern and component directions would likely result in a larger CP. Is this the case? More data shall be presented.

We apologize for the lack of detail in our original submission regarding the exact stimulus parameters in each experiment. We now include these stimulus details in the revised Methods section (p. 24, lines 708-710), a bar graph of the number of sessions that each inter-grating angle was used for each monkey (Supp. Fig. 1b), and a count of the number of neurons tested with each inter-grating angle (Supp. Fig. 2b), as well as the distribution of neuronal preferred directions in these samples (Supp. Fig. 2b). Additionally, we also examined the effect of inter-grating angle on both relative neuronal sensitivity (see our response to the Reviewer's point #4 below for further details) to our textured plaid patterns (Supp. Fig. 4) and choice probability (p. 9-10, lines 267-273 and Fig. 6h). We did not observe a significant effect of inter-grating angle on either metric in either monkey, and this could not be explained by a biased sampling of units with particular preferred

directions. However, sample sizes for many inter-grating angles were small and we agree that a more focused examination of this relationship, with appropriate power, should be a central focus of future studies of this topic.

3. On page 5, lines 136-137, it is stated that “monkey’s responses on zero texture contrast were significantly affected by changes in inter-grating angle”. This is an important validation. Is this result shown anywhere in the paper?

We agree that this is an important claim in our original submission and apologize for the lack of clarity here. In our original report we followed this statement with the details of a regression analysis of performance at zero texture contrast for each monkey versus plaid inter-grating angle but did not include separate plots of this effect. We now provide these plots, as well as example psychometrics from each monkey at three different inter-grating angles, in Supplementary Figure 1 of our revised manuscript.

4. The authors described the dependency of neuronal sensitivity on neuron’s preferred direction relative to the stimulus direction(s) (lines 190-209 on page 7). I suggest to include a data figure showing neuronal sensitivity as a function of the relationship between the preferred direction and the stimulus direction(s).

In our revised manuscript, we now provide separate plots for each monkey illustrating the dependency of neuronal sensitivity on the relationship between a neuron’s preferred direction and the direction(s) of stimulus motion. Specifically, we have added a plot of *relative* neuronal sensitivity, operationalized as threshold ratio (worst pattern direction/best pattern direction), versus normalized preferred direction (best pattern direction – neuronal preferred direction) to the new Figure 5b. Note that here, the “worst”/“best” distinction does not refer to a unit’s sensitivity to the plaid texture cue (i.e., the inverse of its neurometric threshold) but rather which pattern direction was closest to the unit’s preferred direction (measured with single gratings). We also now include a further breakdown of this effect by inter-grating angle (Supplementary Fig. 2a), and (iii) polar plots of the distributions of neuronal preferred directions in each sample (new Fig. 5c and Supp. Fig. 2b).

5. Following point #3, the authors showed the relationship between CP and neuronal sensitivity (Fig. 5c, f), but did not show directly the relationship between CP and relative direction preference, which could be informative.

Thank you, we agree this point was made indirectly through the demonstration of: (i) a dependence of neuronal sensitivity on the pattern direction/preferred direction relationship, and (ii) a correlation between neuronal sensitivity and choice probability. We have followed the reviewer’s instructions and now include this analysis in our revised Results (p. 9-10, lines 267 – 273 and Fig. 6g).

6. The relationship between CP and the pattern index shown in Figure 6d is fairly clear. However, a potential confounding factor is the neuron’s preferred direction relative to the stimulus direction(s). One possibility is that perhaps the pattern cells in the data sample happen to have their preferred directions better aligned with the pattern direction of the

stimulus. Does the trend (CP is positively correlated with the pattern index) still hold if one looks at the neuronal responses to upward direction and downward direction separately (using partial correlation to control for sensitivity)?

Following the reviewers' suggestion we also now tested for differences in the distribution of preferred directions across pattern-component categories (we did not observe a significant difference in the distributions) and include histograms of the neuronal preferred directions in each of the three pattern-component categories (Supp. Fig. 4).

7. Furthermore, does the trend between CP and pattern index still hold if only the CP from zero texture contrast is used?

We thank the reviewer for raising this issue. We originally included only a report of the CP-pattern index relationship using the “grand” CP (calculated from z-scored rates for low contrast (from -20% to +20%) conditions that met our inclusion criterion) as this aggregation procedure has previously been shown to increase the reliability of the CP metric (c.f., Britten et al (1996). *Visual Neuroscience*). Additionally, as is reflected both in our animals' psychometric data, and in subjective reports from human subjects, low contrast plaid stimuli of the type we used *are* variably perceived, e.g., bi-stable, from trial-to-trial. However, to provide an explicit demonstration that the CP-pattern index relationship is consistent regardless of whether or not a perceptually bi-stable stimulus contains clues to surface configuration, we now include an analysis of the CP-pattern index relationship using only the zero texture contrast condition to calculate CP. We now report this correlation in the revised Results (p. 11, lines 306 – 308).

8. Are the regression lines shown in Figure 5c, f, and Figure 6d from type I or II regression? Type II regression should be used if not already.

This is an important point and one that we overlooked in our original submission. We have re-done these analyses using type II regression (geometric mean regression) and updated the statistics in the text.

Minor Comments

1. For the two example neurons shown in Figure 6, indicate their pattern indices and CPs or mark the corresponding data points in the scatter plots of Figure 6c and d.

Thank you, we apologize for the oversight. We have now marked the data points from these neurons in the scatter plots, which have become Figure 7c –d in the revised manuscript.

2. It would be helpful for readers to see the visual stimuli used in this study. Perhaps include a link to the movies.

We agree with the Reviewer that a short movie of the textured plaid stimuli would help readers better understand both our stimulus manipulations and the striking differences in perceptual segmentation that these stimuli engender. Accordingly, we have included

with our revised manuscript a short movie that concatenates drifting plaids of various texture contrast (from coherent texture to non-textured to transparent texture).

3. The papers by McDonald, Clifford, Solomon, Chen, Solomon (2014) in the Journal of Neurophysiology and Xiao and Huang (2015) in the Journal of Neuroscience both looked at the relationship between MT neurons' direction tuning properties relevant to motion segmentation and pattern/component selectivity to plaid stimuli. It may be useful to compare and contrast the current finding with these previous results.

We thank the Reviewer for highlighting these important studies and their relevance to the current work. We apologize for the oversight in our initial submission. We have added mention of the primary findings from these sets of experiments to the revised Introduction (p.2 lines 55-57, and p. 3, lines 71 -76) and situated our findings with respect to these earlier reports in the revised Discussion (p. 16, lines 457-467).

4. How many neurons (units) were recorded and included in the data analysis? How many from monkeys N and S, respectively?

We have now included the total sample sizes in the revised Methods (p. 26-27, lines 781-792).

5. What is the range of the stimulus diameter?

We have now included this information in the revised Methods (p. 26-27, lines 781-792)

6. What is the range of the RF eccentricity?

We have now included this information in the revised Methods (p. 26-27, lines 781-792)

7. What are the luminance values for Lmax and Lmin at 80% and 0% Michelson contrast?

Texture contrast luminance was defined relative to grating luminance. This information is now provided in the revised Methods (p. 24, lines 722-729).

Reviewer #2

This paper aims to show that Area MT in monkeys has a direct correlation with the perception of motion. Several authors have already demonstrated that MT appears to be involved in 'driving' the perception of motion. The specific issue investigated here is whether MT (and particularly one cell type in MT) is involved in the perception of motion segmentation judgements. The most important finding is that a subset of cells in Area MT, which have been labelled pattern cells based on a single test, is particularly correlated with segmentation.

The work is certainly of interest to people in the field. However, as the authors mention themselves, one of the largest problems in my mind is the importance of the finding in a broader context. Pattern and Component cells are defined based on a very restricted stimulus-type: namely plaid patterns. Pattern and component cells do not form two distinct groups but, rather are defined based on a statistical test that splits cells into three groups that do not naturally split into three separate clusters. There is no border between

these groups (i.e. statistics don't allow you to identify three groups with dips between them). In truth, there is a continuum and most cells are a bit of both extremes. Plaid patterns contain restricted amounts of information on motion direction, so the lack of clear boundaries is not surprising. Therefore, saying that pattern cells are more closely linked to the perception of transparent motion appears philosophically to be a rather small increment. For example, what would happen if more certainty were built into the stimulus, such as occurs immediately, perceptually, when terminators appear in the RF? Would cells defined as pattern-selective differ from the other MT cells in this case and would this influence the bias of the single-cell responses? Would the 'special' case for pattern cells persist?

The work was conducted in behaving primates and this is a very challenging and time consuming technique. To the best of my knowledge the work was conducted at the highest level and the statistics are appropriate. I believe that the experiments could be repeated by following the methods. I am not in a position to analyse the data to see if I can reproduce the same result.

Therefore, while the work is of the highest quality, the paper suffers in my view from not clearly showing how the MT population code as a whole deals with transparency and segmentation judgements. I feel that a model that incorporates all of the response types in MT would be essential in trying to answer this. The authors could develop a data-driven model that uses their hard-won monkey data. As it stands, I feel that the paper attempts to highlight a correlation between perception and just one type of MT cell (i.e. pattern cells).

We would like to thank the Reviewer for acknowledging the challenging nature of these types of experiments. We appreciate their positive assessment of the quality and potential impact of the work.

We agree with the reviewer that further studies, building upon our findings here, will be necessary to tease apart:

(i) The general role of MT pattern- and component-cell populations in the perception of diverse types of visual motion stimuli.

and

(ii) The circuit-level mechanisms by which a population code for the perceptual segmentation of visual motion signals is implemented in the primate motion processing pathway.

We thank the reviewer for raising these important concerns. Below, we outline, in turn, how we have addressed each of these points in our revised manuscript:

(i) The Reviewer posits: "...[w]hat would happen if more certainty were built into the stimulus, such as occurs immediately, perceptually, when terminators appear in the RF? Would cells defined as pattern-selective differ from the other MT cells in this case and

would this influence the bias of the single-cell responses? Would the 'special' case for pattern cells persist?"

This is a difficult hypothetical to address. First, we would take care to note that, as demonstrated by Kooi (1993, *Vision Research*), perception of some stimuli that include both ambiguous (oriented contour) and unambiguous (terminator) motion signals does not immediately track the pattern direction. Instead, observers initially report that these stimuli move in the contour direction before appearing to move in the terminator direction. The timescale and magnitude of this shift depends, amongst other factors, upon the relative strength of ambiguous and unambiguous signals in the display (cf. Shimojo et al (1998). *Vision Research*; Fisher & Zanker (2001). *Perception*). Second, neuronal signaling, in both V1 (Pack et al (2003). *Neuron*) and MT (Pack et al (2004). *J. Neurosci.*), of terminator motion in these types of stimuli similarly lags contour motion signaling. Third, in the above comment, the reviewer does not specify a particular psychophysical task – e.g., fine or coarse discrimination of differences in terminator motion? – and the relationship between choice probability (CP) and neuronal tuning is well known to be affected by task (cf. Purushothaman & Bradley (2005). *Nature Neuroscience*; and Chowdhury & DeAngelis (2008). *Neuron*, for examples in MT). This relationship between task structure and CP is one we discussed in our initial submission. However, even given these caveats, we agree with what we understand to be the reviewer's overarching point, namely, that the generalizability of our finding of a significant correlation between CP and pattern index is an important question that warrants further investigation. Accordingly, we have toned down all strong claims regarding the CP-PI link and amended our manuscript with further discussion along these lines. Specifically:

1. To address what we believe is the Reviewer's general point, we now explicitly include discussion of the need for future studies of the role of MT activity in perception to address directly differences in pattern- and component-cell choice probabilities regardless of stimulus or task (p. 15, lines 426-432 and p.15 440-445). Visual motion signals are used for a number of computations, e.g., ocular following, heading perception, perceptual segmentation, etc. it is possible that what the Reviewer terms the "special" case for pattern cells that we observe in our segmentation task might not persist for tasks that do not require image segmentation, such as detecting a signal in noise or tracking a signal moving object.

(ii) The Reviewer states: "...[t]he paper suffers in my view from not clearly showing how the MT population code as a whole deals with transparency and segmentation judgements"

And, finally, the Reviewer suggests: "...[T]he authors could develop a data-driven model that uses their hard-won monkey data. As it stands, I feel that the paper attempts to highlight a correlation between perception and just one type of MT cell (i.e. pattern cells)."

We thank the reviewer for challenging us to think hard about what we can conclude about the MT population code for segmentation judgments based on our findings.

There is currently an ongoing debate within the literature regarding the generation of choice signals in sensory cortex, with both top-down (cf Nienborg et al (2012). *Ann Rev Neuro* for review) and bottom-up (cf Hafner et al (2018). *Neuron*, for an example) models accounting for similar phenomena. Similarly, there is continued disagreement as to how MT pattern and component receptive fields are constructed.

Given the above, we feel that a computational model that provides a real contribution to our understanding of how the MT population code is organized by both: (i) accounting for both the findings we report in our manuscript as well as historical data, and (ii) providing testable predictions about differences in pattern- and component-cell choice probabilities for arbitrary stimuli goes far beyond the bounds of the current study. However, although we do not feel we have sufficient data to construct, or space for including here, such a formal model we do feel that we can provide an informal illustration of different categories of circuits that could underlie our observations. We hope that this will help in suggesting some future studies that could attempt to rule out one or more of these potential alternatives. Additionally, we believe this will aid in developing a formal model in a future stand-alone study. In our revision, we provide this schematic in our new Figure 8 and discuss these points in the associated figure caption (referenced in the Discussion, p. 16, line 479).

Reviewer #3

This has been one of the easiest papers I have reviewed. The authors have done an excellent job addressing an open question about the role of area MT in discriminating between transparent versus coherent motion. Results show that MT responses signal transparent versus coherent motion, are correlated with behavior, and are stronger in pattern direction cells, the cells that were previously hypothesized to play a more direct role in motion perception.

The experiment is well designed and potential shortcomings are properly discussed. The data analysis follows classic work, is thorough, and acknowledges open questions. Finally, the conclusions are consistent with the observations, and the potential caveats are properly discussed.

First, we would like to express our gratitude for Dr. Jazayeri's positive comments on our initial submission.

I only have one comment that I would like the authors to discuss. When looking at transparent versus coherent motion stimuli, one gets the feeling that the two are very different perceptually; intuitively, the effect doesn't seem to boil down to a few spikes more or less in one direction or another. Do the authors really believe that a just-above-chance CP is all it takes to flip the perception qualitatively? I know this type of thinking is not common in classical work in area MT, but I think it is useful for the readers to hear the authors' speculation on how such discrete precepts might emerge from small graded differences in firing rates.

Otherwise, a nice piece of work

-Mehrdad Jazayeri

We agree that the question of how such sharp differences in perception could arise from graded differences in MT firing rates is a critical one. We appreciate Dr. Jazayeri's suggestion that we add some speculation regarding this point. We have addressed this request in three ways:

1. First, to make sure that Dr. Jazayeri's point regarding the subjective quality of coherent and transparent percepts is clear for readers unfamiliar with these stimuli, we have included with our revised submission a short supplementary movie file depicting the perceptual transformation in our plaids from clearly transparent to clearly coherent motion with changes in the strength of the texturing cue.

2. One clear possibility is that we underestimated the magnitude of the choice effect in MT. For example, by integrating over the entire 1.5s stimulus presentation period to calculate CP. Given that it is now well established that MT pattern motion signaling, in conventional plaids (Smith et al (2005). *Nature Neuroscience*) and other stimuli (Pack & Born (2001). *Nature*; Pack et al (2004). *J. Neurosci.*; Jazayeri et al (2011). *J. Neurosci.*), takes time to develop, we reasoned that including the early portion of the response in our calculation could be diluting the choice effect in MT. Calculating CP in sliding windows throughout the trial confirmed that this was indeed the case. This new analysis is now presented in the new Figure 6i and discussed in the Results (p. 10, lines 274-282). In our Discussion (p. 13-14, lines 383-403), we also acknowledge the possibility that a reaction-time rather than fixed viewing duration design, in which the animal was free to indicate its perception at any point in the trial, could have resulted in a larger choice effect.

3. Finally, we now include some speculation as to how different pooling models, such as those that have been previously implemented to account for how small graded changes in MSTd firing rates drive decisions about heading direction (cf. Gu et al (2014). *Elife*), could possibly account for our findings of small graded difference in MT firing rates correlating with sharp differences in subjective perception (p. 14, lines 403-414).

REVIEWERS' COMMENTS

Reviewer #1 (Remarks to the Author):

The authors have done a thorough job in responding to my (and other reviewers') previous comments. The revision of the manuscript and the additional data and analysis results provided have improved this paper significantly. I have no further comments.

Reviewer #2 (Remarks to the Author):

I believe that the authors have addressed my deliberately philosophical issues well. I am happy with the additions made. They were not in a position to directly tackle my questions without further research but I think this is unnecessary as they have addressed the issues as far as possible at this stage. The important thing was for them to note these issues in the text.